# Strategic Classification with Graph Neural Networks

**Itay Eilat**[*]**, Ben Finkelshtein**[*]**, Chaim Baskin, Nir Rosenfeld**
Technion – Israel Institute of Technology
{itayeilat,benfin}@campus.technion.ac.il
{chaimbaskin,nirr}@cs.technion.ac.il

## Abstract

Strategic classification studies learning in settings where users can modify their features to obtain favorable predictions. Most current works focus on simple classifiers that trigger independent user responses. Here we examine the implications of learning with more elaborate models that break the independence assumption. Motivated by the idea that applications of strategic classification are often social in nature, we focus on *graph neural networks*, which make use of social relations between users to improve predictions. Using a graph for learning introduces inter-user dependencies in prediction; our key point is that strategic users can exploit these to promote their own goals. As we show through analysis and simulation, this can work either against the system—or for it. Based on this, we propose a differentiable framework for strategically-robust learning of graph-based classifiers. Experiments on several real networked datasets demonstrate the utility of our approach.

## 1 Introduction

Machine learning is increasingly being used to inform decisions about humans. But when users of a system stand to gain from certain predictive outcomes, they may be prone to "game" the system by strategically modifying their features (at some cost). The literature on *strategic classification* (Brückner & Scheffer, 2011; Hardt et al., 2016) studies learning in this setting, with emphasis on how to learn classifiers that are robust to strategic user behavior. The idea that users may respond to a decision rule applies broadly and across many domains, from hiring, admissions, and scholarships to loan approval, insurance, welfare benefits, and medical eligibility (McCrary, 2008; Almond et al., 2010; Camacho & Conover, 2011; Lee & Lemieux, 2010). This, along with its clean formulation as a learning problem, have made strategic classification the target of much recent interest (Sundaram et al., 2021; Zhang & Conitzer, 2021; Levanon & Rosenfeld, 2021; Ghalme et al., 2021; Jagadeesan et al., 2021; Zrnic et al., 2021; Estornell et al., 2021; Lechner & Urner, 2021; Harris et al., 2021; Levanon & Rosenfeld, 2022; Liu et al., 2022; Ahmadi et al., 2022; Barsotti et al., 2022a).

But despite these advances, most works in strategic classification remain to follow the original problem formulation in assuming independence across users responses. From a technical perspective, this assumption greatly simplifies the learning task, as it allows the classifier to consider each user's response in isolation: user behavior is modeled via a *response mapping* $\Delta_h(x)$ determining how users modify their features $x$ in response to the classifier $h$, and learning aims to find an $h$ for which $y \approx h(\Delta_h(x))$. Intuitively, a user will modify her features if this 'moves' her across the decision boundary, as long as this is worthwhile (i.e., gains from prediction exceed modification costs). Knowing $\Delta_h$ allows the system to anticipate user responses and learn an $h$ that is robust.

For a wide range of settings, learning under independent user responses has been shown to be theoretically possible (Hardt et al., 2016; Zhang & Conitzer, 2021; Sundaram et al., 2021) and practically feasible (Levanon & Rosenfeld, 2021; 2022). Unfortunately, once this assumption of independence is removed—results no longer hold. One reason is that current approaches can safely assume independence because the decision rules they consider *induce* independence: when predictions inform decisions for each user independently, users have no incentive to account for the behavior of others. This limits the scope of predictive models to include only simple functions of single inputs.

---

[*]Equal contribution, alphabetical order

In this paper, we aim to extend the literature on strategic classification to support richer learning paradigms that enable inter-dependent user responses, with particular focus on the domain of Graph Neural Networks (GNNs) (Monti et al., 2017; Wang et al., 2019; Bronstein et al., 2017; Hamilton et al., 2017). Generally, user responses can become dependent through the classifier if predictions for one user rely also on information regarding other users, i.e., if $h(x_i)$ is also a function of other $x_j$. In this way, the affects of a user modifying her features via $x_j \mapsto \Delta_h(x_j)$ can propagate to other users and affect their decisions (since $h(x_i)$ now relies on $\Delta_h(x_j)$ rather than $x_j$). For GNNS, this expresses through their relience on the graph. GNNs take as input a weighted graph whose nodes correspond to featurized examples, and whose edges indicate relations that are believed to be useful for prediction (e.g., if $j{\rightarrow}i$ indicates that $y_i = y_j$ is likely). In our case, nodes represent users, and edges represent social links. The conventional approach is to first embed nodes in a way that depends on their neighbors' features, $\phi_i = \phi(x_i; x_{\mathrm{nei}(i)})$, and then perform classification (typically linear) in embedded space, $\hat{y}_i = \mathrm{sign}(w^\top \phi_i)$. Notice $\hat{y}_i$ depends on $x_i$, but also on all other $x_j \in x_{\mathrm{nei}(i)}$; hence, in deciding how to respond, user $i$ must also account for the strategic responses of her neighbors $j \in \mathrm{nei}(i)$. We aim to establish the affects of such dependencies on learning.

As a concrete example, consider Lenddo[1], a company that provides credit scoring services to lending institutions. Lenddo specializes in consumer-focused microlending for emerging economies, where many applicants lack credible financial records. To circumvent the need to rely on historical records, Lenddo uses applicants' social connections, which are easier to obtain, as a factor in their scoring system.[2] As an algorithmic approach for this task, GNNs are an adequate choice (Gao et al., 2021). Once loan decisions become dependent on social relations, the incentives for acting strategically change (Wei et al., 2016). To see how, consider that a user who lies far to the negative side of the decision boundary (and so independently cannot cross) may benefit from the graph if her neighbors "pull" her embedding towards the decision boundary and close enough for her to cross. Conversely, the graph can also suppress strategic behavior, since neighbors can "hold back" nodes and prevent them from crossing. Whether this is helpful to the system or not depends on the true label of the node.

This presents a tradeoff: In general, graphs are useful if they are informative of labels in a way that complements features; the many success stories of GNNs suggest that this is often the case (Zhou et al., 2020). But even if this holds *sans* strategic behavior—once introduced, graphs inadvertently create dependencies through user representations, which strategic users can exploit. Graphs therefore hold the potential to benefit the system, but also its users. Here we study the natural question: who does the graph help more? Through analysis and experimentation, we show that learning in a way that *neglects* to account for strategic behavior not only jeopardizes performance, but becomes *worse* as reliance on the graph increases. In this sense, the graph becomes a vulnerability which users can utilize for their needs, turning it from an asset to the system—to a potential threat.

As a solution, we propose a practical approach to learning GNNs in strategic environments. We show that for a key neural architecture (SGC; Wu et al. (2019)) and certain cost functions, graph-dependent user responses can be expressed as a 'projection-like' operator. This operator admits a simple and differentiable closed form; with additional smoothing, this allows us to implement responses as a neural layer, and learn robust predictors $h$ using gradient methods. Experiments on synthetic and real data (with simulated responses) demonstrate that our approach not only effectively accounts for strategic behavior, but in some cases, can harness the efforts of self-interested users to promote the system's goals. Our code is publicly available at: http://github.com/StrategicGNNs/Code.

## 1.1 RELATED WORK

**Strategic classification.** Since its introduction in Hardt et al. (2016) (and based on earlier formulations in Brückner & Scheffer (2009); Brückner et al. (2012); Großhans et al. (2013)), the literature on strategic classification has been growing at a rapid pace. Various aspects of learning have been studied, including: generalization behavior (Zhang & Conitzer, 2021; Sundaram et al., 2021; Ghalme et al., 2021), algorithmic hardness (Hardt et al., 2016), practical optimization methods (Levanon & Rosenfeld, 2021; 2022), and societal implications (Milli et al., 2019; Hu et al., 2019; Chen et al., 2020; Levanon & Rosenfeld, 2021). Some efforts have been made to extend beyond the conventional user models, e.g., by adding noise (Jagadeesan et al., 2021), relying on partial information (Ghalme

---

[1] http://lenddoefl.com; see also http://www.wired.com/2014/05/lenddo-facebook/.
[2] For a discussion on ethics, see final section. For similar initiatives, see https://en.wikipedia.org/wiki/Lenddo.

et al., 2021; Bechavod et al., 2022), or considering broader user interests (Levanon & Rosenfeld, 2022); but these, as do the vast majority of other works, focus on linear classifiers and independent user responses.[3] We study richer predictive model classes that lead to correlated user behavior.

**Graph Neural Networks (GNNs).** The use of graphs in learning has a long and rich history, and remains a highly active area of research (Wu et al., 2020). Here we cover a small subset of relevant work. The key idea underlying most methods is to iteratively propagate and aggregate information from neighboring nodes. Modern approaches implement variations of this idea as differentiable neural architectures (Gori et al., 2005; Scarselli et al., 2008; Kipf & Welling, 2017; Gilmer et al., 2017). This allows to express more elaborate forms of propagation (Li et al., 2018; Alon & Yahav, 2021) and aggregation (Wu et al., 2019; Xu et al., 2019; Li et al., 2016), including attention-based mechanisms (Veličković et al., 2018; Brody et al., 2022). Nonetheless, a key result by Wu et al. (2019) shows that, both theoretically and empirically, linear GNNs are also quite expressive.

**Robustness of GNNs.** As most other fields in deep learning, GNNs have been the target of recent inquiry as to their sensitivity to adversarial attacks. Common attacks include perturbing nodes, either in sets (Zügner et al., 2018; Zang et al., 2021) or individually (Finkelshtein et al., 2020). Attacks can be applied before training (Zügner & Günnemann, 2019; Bojchevski & Günnemann, 2019; Li et al., 2021; Zhang & Zitnik, 2020) or at test-time (Szegedy et al., 2014; Goodfellow et al., 2015); our work corresponds to the latter. While there are connections between adversarial and strategic behavior (Sundaram et al., 2021), the key difference is that strategic behavior is not a zero-sum game; in some cases, incentives can even align (Levanon & Rosenfeld, 2022). Thus, system-user relations become more nuanced, and provide a degree of freedom in learning that does not exist in adversarial settings.

## 2 LEARNING SETUP

Our setting includes $n$ users, represented as nodes in a directed graph $G = (V, E)$ with non-negative edge weights $W = \{w_{ij}\}_{(i,j)\in E}, w_{ij} \geq 0$. Each user $i$ is also described by a feature vector $x_i \in \mathbb{R}^\ell$ and a binary label $y_i \in \{\pm 1\}$. We use $x_{-i} = \{x_j\}_{j\neq i}$ to denote the set of features of all nodes other than $i$. Using the graph, our goal is to learn a classifier $h$ that correctly predicts user labels. The challenge in our strategic setting is that inputs at test-time can be strategically modified by users, in response to $h$ and in a way that depends on the graph and on other users (we describe this shortly). Denoting by $x_i^h$ the (possibly modified) strategic response of $i$ to $h$, our learning objective is:

$$\underset{h\in H}{\text{argmin}} \sum_i L(y_i, \hat{y}_i), \qquad \hat{y}_i = h(x_i^h; x_{-i}^h) \qquad (1)$$

where $H$ is the model class and $L$ is a loss function (i.e., log-loss). Note that both predictions $\hat{y}_i$ and modified features $x_i^h$ can depend on $G$ and on on $x_{-i}^h$ (possibly indirectly through $h$). We focus on the inductive graph learning setting, in which training is done on $G$, but testing is done on a different graph, $G'$ (often $G, G'$ are two disjoint components of a larger graph). Our goal is therefore to learn a classifier that generalizes to other graphs in a way that is robust to strategic user behavior.

**Graph-based learning.** We consider linear graph-based classifiers—these are linear classifiers that operate on linear, graph-dependent node embeddings, defined as:

$$h_{\theta,b}(x_i; x_{-i}) = \text{sign}(\theta^\top \phi(x_i; x_{-i}) + b), \qquad \phi(x_i; x_{-i}) = \widetilde{w}_{ii}x_i + \sum_{j\neq i} \widetilde{w}_{ji}x_j \qquad (2)$$

where $\phi_i = \phi(x_i; x_{-i})$ is node $i$'s embedding,[4] $\theta \in \mathbb{R}^\ell$ and $b \in \mathbb{R}$ are learned parameters, and $\widetilde{w}_{ij} \geq 0$ are pairwise weights that depend on $G$ and $W$. We refer to users $j$ with $\widetilde{w}_{ji} \neq 0$ as the *embedding neighbors* of $i$. A simple choice of weights is $\widetilde{w}_{ji} = w_{ji}$ for $(j, i) \in E$ (and 0 otherwise), but different methods propose different ways to construct $\widetilde{w}$; here we adopt the weight scheme of Wu et al. (2019). We assume the weights $\widetilde{w}$ are predetermined, and aim to learn $\theta$ and $b$ in Eq. (1).

Our focus on linear GNNs stems from several factors. From the perspective of strategic classification, linear decision rules ensure that strategic responses are computationally tractable (see Eq. (4)). This is conventionally required, and most works remain in the linear regime. From the perspective of GNNs,

---

[3]The only exception we know of is Liu et al. (2022) who study strategic ranking, but do not consider learning.
[4]Note that embedding preserve the dimension of the original features.

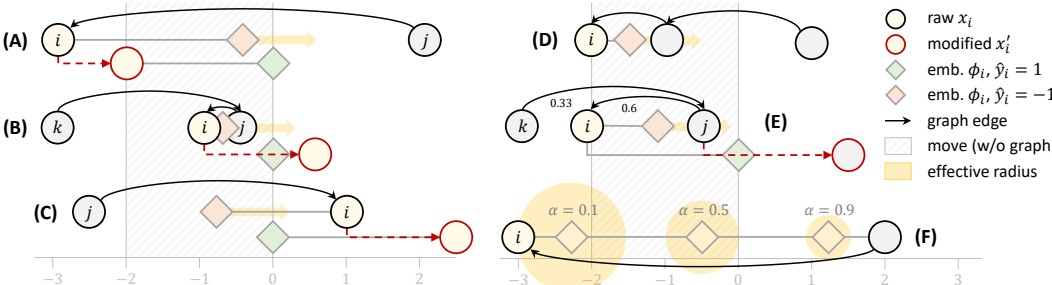

Figure 1: **Simple graphs, complex behavior**. In all examples $h_b$ has $b = 0$, and $\widetilde{w}_{ij} = 1$ unless otherwise noted. **(A)** Without the graph, $i$ does not move, but with $j{\to}i$, $\phi_i$ is close enough to cross. **(B)** $i$ can cross, but due to $j{\to}i$, must move beyond 0. **(C)** Without the graph, $i$ has $\hat{y}_i = 1$, but due to $j{\to}i$, must move. **(D)** Without the graph $i$ would move, but with the graph—cannot. **(E)** Hitchhiker: $i$ cannot move, but once $j$ moves and crosses, $\phi_i$ crosses for free. **(F)** Effective radii for various $\alpha$.

linear architectures have been shown to match state-of-the-art performance on multiple tasks (Wu et al., 2019), implying sufficiently manifest the fundamental role of graphs. Thus, linear GNNs serve as a minimal necessary step for bridging standard strategic classification and graph-based learning, in a way that captures the fundamental structure of the learning task in both domains. Nonetheless, as we show in Sec. 4, even for linear GNNs—user responses can cause learning to be highly non-linear.

**Strategic inputs.** For the strategic aspects of our setting, we build on the popular formulation of Hardt et al. (2016). Users seek to be classified positively (i.e., have $\hat{y}_i = 1$), and to achieve this, are willing to modify their features (at some cost). Once the system has learned and published $h$, a test-time user $i$ can modify her features $x_i \mapsto x_i'$ in response to $h$. Modification costs are defined by a cost function $c(x, x')$ (known to all); here we focus mainly on 2-norm costs $c(x, x') = \|x - x'\|_2$ (Levanon & Rosenfeld, 2022; Chen et al., 2020), but also discuss other costs (Brückner et al., 2012; Levanon & Rosenfeld, 2021; Bechavod et al., 2022). User $i$ modifies her features (or "moves") if this improves her prediction (i.e., if $h(x_i) = -1$ but $h(x_i') = 1$) and is cost-effective (i.e., prediction gains exceed modification costs); for linear classifiers, this means crossing the decision boundary. Note that since $y \in \{\pm 1\}$, gains are at most $h(x') - h(x) = 2$. Users therefore do not move to any $x'$ whose cost $c(x, x')$ exceeds a 'budget' of 2, and the maximal moving distance is $d = 2$.

**Distribution shift.** One interpretation of strategic classification is that user responses cause *distribution shift*, since in aggregate, $p(x') \neq p(x)$. Crucially, *how* the distribution changes depends on $h$, which implies that the system has some control over the test distribution $p(x')$, indirectly through user how users respond—a special case of model-induced distribution shift (Miller et al., 2021; Maheshwari et al., 2022). The unique aspect of our setting is that user responses are linked through their mutual dependence on the graph. We next describe our model of user responses in detail.

## 3 STRATEGIC USER BEHAVIOR: MODEL AND ANALYSIS

Eq. (2) states that $h$ classifies $i$ according to her embedding $\phi_i$, which in turn is a weighted sum of her features and those of her neighbors. To gain intuition as to the effects of the graph on user behavior, it will be convenient to assume weights $\widetilde{w}$ are normalized[5] so that we can write:

$$\phi_i = \phi(x_i; x_{-i}) = (1 - \alpha_i)x_i + \alpha_i \bar{x}_i \quad \text{for some} \ \alpha_i \in [0, 1] \tag{3}$$

I.e., $\phi_i$ can be viewed as an interpolation between $x_i$ and some point $\bar{x}_i \in \mathbb{R}^\ell$ representing all other nodes, where the precise point along the line depends on a parameter $\alpha_i$ that represents the influence of the graph (in a graph-free setting, $\alpha_i = 0$). This reveals the dual effect a graph has on users: On the one hand, the graph limits the ability of user $i$ to influence her own embedding, since any effort invested in modifying $x_i$ affects $\phi_i$ by at most $1 - \alpha_i$. But the flip side of this is that an $\alpha_i$-portion of $\phi_i$ is fully determined by other users (as expressed in $\bar{x}_i$); if they move, $i$'s embedding also 'moves' for free. A user's 'effective' movement radius is $r_i = d(1 - \alpha_i)$. Fig. 1 (F) shows this for varying $\alpha_i$.

---

[5]This is indeed the case in several common approaches.

### 3.1 STRATEGIC RESPONSES

Given that $h$ relies on the graph for predictions—how should a user modify her features $x_i$ to obtain $\hat{y}_i = 1$? In vanilla strategic classification (where $h$ operates on each $x_i$ independently), users are modeled as rational agents that respond to the classifier by maximizing their utility, i.e., play $x_i' = \text{argmax}_{x'} h(x') - c(x_i, x')$, which is a best-response that results in immediate equilibrium (users have no incentive to move, and the system has no incentive to change $h$).[6] In our graph-based setting, however, the dependence of $\hat{y}_i$ on all other users via $h(x_i; x_{-i})$ makes this notion of best-response ill-defined, since the optimal $x_i'$ can depend on others' strategic responses, $x_{-i}'$, which are unknown to user $i$ at the time of decision (and may very well rely on $x_i'$ itself).

As a feasible alternative, here we generalize the standard model by assuming that users play *myopic best-response* over a sequence of multiple update rounds. As we will see, this has direct connections to key ideas underlying graph neural networks. Denote the features of node $i$ at round $t$ by $x_i^{(t)}$, and set $x_i^{(0)} = x_i$. A myopic best response means that at round $t$, each user $i$ chooses $x_i^{(t)}$ to maximize her utility at time $t$ *according to the state of the game at time $t-1$*, i.e., assuming all other users play $\{x_j^{(t-1)}\}_{j \neq i}$, with costs accumulating over rounds. This defines a *myopic response mapping*:

$$\Delta_h(x_i; x_{-i}, \kappa) \triangleq \underset{x' \in \mathbb{R}^\ell}{\text{argmax}} \, h(x'; x_{-i}) - c(x_i, x') - \kappa \tag{4}$$

where at round $t$ updates are made (concurrently) via $x_i^{(t+1)} = \Delta_h(x_i^{(t)}; x_{-i}^{(t)}, \kappa_i^{(t)})$ with accumulating costs $\kappa_i^{(t)} = \kappa_i^{(t-1)} + c(x_i^{(t-1)}, x_i^{(t)})$, $\kappa_i^{(0)} = 0$. Predictions for round $t$ are $\hat{y}_i^{(t)} = h(x_i^{(t)}; x_{-i}^{(t)})$.

Eq. (4) naturally extends the standard best-response mapping (which is recovered when $\alpha_i = 0 \, \forall i$, and converges after one round). By adding a temporal dimension, the actions of users propagate over the graph and in time to affect others. Nonetheless, even within a single round, graph-induced dependencies can result in non-trivial behavior; some examples for $\ell = 1$ are given in Fig. 1 (A-D).

### 3.2 ANALYSIS

We now give several results demonstrating basic properties of our response model and consequent dynamics, which shed light on how the graph differentially affects the system and its users.

**Convergence.** Although users are free to move at will, movement adheres to a certain useful pattern.

**Proposition 1.** *For any $h$, if users move via Eq. (4), then for all $i \in [n]$, $x_i^{(t)} \neq x_i^{(t-1)}$ at most once.*

*Proof.* User $i$ will move only when: (i) she is currently classified negatively, $h(x_i; x_{i-}) = -1$, and (ii) there is some $x'$ for which utility can improve, i.e., $h(x'; x_{i-}) - c(x_i, x') > -1$, which in our case occurs if $h(x'; x_{i-}) = 1$ and $c(x_i, x') < 2$ (since $h$ maps to $[-1, 1]$).[7] Eq. (4) ensures that the modified $x_i'$ will be such that $\phi(x_i'; x_{-i})$ lies exactly on the decision boundary of $h$; hence, $x_i'$ must be *closer* to the decision boundary (in Euclidian distance) than $x_i$. This means that any future moves of an (incoming) neighbor $j$ can only push $i$ further away from the decision boundary; hence, the prediction for $i$ remains positive, and she has no future incentive to move again.[8] $\square$

Hence, all users move at most once. The proof reveals a certain monotonicity principle: users always (weakly) benefit from any strategic movement of others. Convergence follows as an immediate result.

**Corollary 1.** *Myopic-best response dynamics converge for any $h$ (and after at most $n$ rounds).*

We will henceforth use $x_i^h$ to denote the features of user $i$ at convergence (w.r.t. $h$), denoted $T_{\max}$.

**Hitchhiking.** When $i$ moves, the embeddings of (outgoing) neighbors $j$ who currently have $\hat{y}_j = -1$ also move closer to the decision boundary; thus, users who were initially too far to cross may be able to

---

[6]Note 'rational' here implies users are assumed to know $h$. As most works in the field, we also make this assumption; for the practically-inclined reader, note that (i) in some cases, there is reason to believe it may approximately hold (e.g., http://openschufa.de), and (ii) relaxing this assumption (and others) is an ongoing community effort (Ghalme et al., 2021; Jagadeesan et al., 2021; Bechavod et al., 2022; Barsotti et al., 2022b).

[7]In line with Hardt et al. (2016), we assume that if the value is zero then the user does not move.

[8]Users moving only once ensures that cumulative costs are never larger than the final gain.

do so at later rounds. In this sense, the dependencies across users introduced by the graph-dependent embeddings align user incentives, and promote an implicit form of cooperation. Interestingly, users can also obtain positive predictions *without* moving. We refer to such users as 'hitchhikers'.

**Proposition 2.** *There exist cases where $\hat{y}_i^{(t)} = -1$ and $i$ doesn't move, but $\hat{y}_i^{(t+1)} = 1$.*

A simple example can be found in Figure 1 (E). Hitchhiking demonstrates how relying on the graph for classification can promote strategic behavior—even under a single response round.

**Cascading behavior.** Hitchhiking shows how the movement of one user can flip the label of another, but the effects of this process are constrained to a single round. When considering multiple rounds, a single node can trigger a 'domino effect' of moves that span the entire sequence.

**Proposition 3.** *For any $n$, there exists a graph where a single move triggers $n$ additional rounds.*

**Proposition 4.** *For any $n$ and $k \leq n$, there exists a graph where $n - k$ users move at round $k$.*

Proofs are constructive and modular, and rely on graphs that are predictively useful (Appendix A.2). Note also that graph diameter is not a mediating factor (Appendix E.3). Both results show that, through monotonicity, users also (weakly) benefit from additional rounds. This has concrete implications.

**Corollary 2.** *In the worst case, the number of rounds until convergence is $\Omega(n)$.*

**Corollary 3.** *In the worst case, $\Omega(n)$ users move after $\Omega(n)$ rounds.*

Thus, to exactly account for user behavior, the system must correctly anticipate the strategic responses of users many rounds into the future, since a bulk of predictions may flip in the last round. Fortunately, these results also suggests that in some cases, blocking one node from crossing can prevent a cascade of flips; thus, it may be worthwhile to 'sacrifice' certain predictions for collateral gains. This presents an interesting tradeoff in learning, encoded in the learning objective we present next, and which we motivate with our final result on the potential impact of strategic behavior:

**Proposition 5.** *The gap in accuracy between (i) the optimal non-strategic classifier on non-strategic data, and (ii) the optimal strategic classifier on strategic data, can be as large as 30% (see Apx. E.1).*

## 4 LEARNING AND OPTIMIZATION

We are now ready to describe our learning approach. Our learning objective can be restated as:

$$\hat{h} = \underset{h \in H}{\operatorname{argmin}} \sum_i L(y_i, h(x_i^h; x_{-i}^h)) \tag{5}$$

for $H = \{h_{\theta,b}\}$ as in Eq. (2). The difficulty in optimizing Eq. (5) is that $x^h$ depend on $h$ through the iterative process, which relies on $\Delta_h$. At test time, $x^h$ can be computed exactly by simulating the dynamics. However, at train time, we would like to allow for gradients of $\theta, b$ to propagate through $x^h$. For this, we propose an efficient differential proxy of $x^h$, implemented as a stack of layers, each corresponding to one response round. The number of layers is a hyperparameter, $T$.

**Single round.** We begin with examining a single iteration of the dynamics, i.e., $T = 1$. Note that since a user moves only if the cost is at most 2, Eq. (4) can be rewritten as:

$$\Delta_h(x_i; x_{-i}) = \begin{cases} x_i' & \text{if } h(x_i; x_{-i}) = -1 \text{ and } c(x_i, x_i') \leq 2 \\ x_i & \text{o.w.} \end{cases} \tag{6}$$

where $x_i' = \operatorname{proj}_h(x_i; x_{-i})$ is the point to which $x_i$ must move in order for $\phi(x_i; x_{-i})$ to be projected onto $h$. This projection-like operator (on $x_i$) can be shown to have a closed-form solution:

$$\operatorname{proj}_h(x_i; x_{-i}) = x_i - \frac{\theta^\top \phi(x_i; x_{-i}) + b}{\|\theta\|_2^2 \widetilde{w}_{ii}} \theta \tag{7}$$

See Appendix B.1 for a derivation using KKT conditions. Eq. (7) is differentiable in $\theta$ and $b$; to make the entire response mapping differentiable, we replace the 'hard if' in Eq. (6) with a 'soft if', which we now describe. First, to account only for negatively-classified points, we ensure that only points in the negative halfspace are projected via a 'positive-only' projection:

$$\operatorname{proj}_h^+(x_i; x_{-i}) = x_i - \min \left\{ 0, \frac{\theta^\top \phi(x_i; x_{-i}) + b}{\|\theta\|_2^2 \widetilde{w}_{ii}} \right\} \theta \tag{8}$$

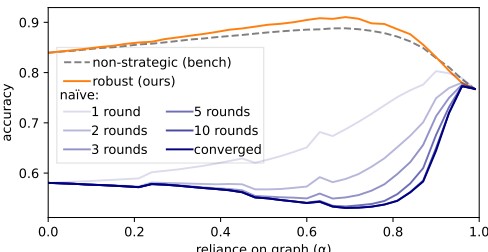 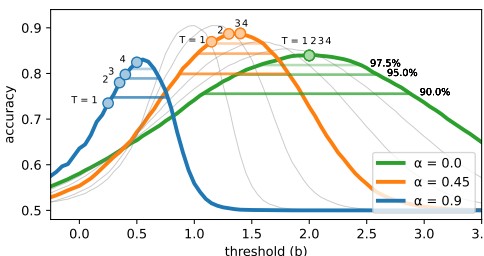

Figure 2: **(Left)** Accuracy of learned $\hat{h}$ for varying graph importances $\alpha \in [0, 1]$. **(Right)** Accuracy for all threshold classifiers $h_b$, with sensitivity (horizontal lines) and per network depth $T$ (dots).

Then, we replace the $c \leq 2$ constraint with a smoothed sigmoid that interpolates between $x_i$ and the projection, as a function of the cost of the projection and thresholded at 2. This gives our differentiable approximation of the response mapping:

$$\tilde{\Delta}(x_i; x_{-i}, \kappa) = x_i + (x'_i - x_i)\sigma_\tau\big(2 - c(x_i, x'_i) - \kappa\big) \quad \text{where} \quad x'_i = \text{proj}_h^+(x_i; x_{-i}) \quad (9)$$

where $\sigma$ is a sigmoid and $\tau$ is a temperature hyperparameter ($\tau \to 0$ recovers Eq. (6)) and for $T = 1$, $\kappa = 0$. In practice we add a small additive tolerance term for numerical stability (See Appendix B.3).

**Multiple rounds.** Next, we consider the computation of (approximate) modified features after $T > 1$ rounds, denoted $\tilde{x}^{(T)}$, in a differentiable manner. Our approach is to apply $\tilde{\Delta}$ iteratively as:

$$\tilde{x}_i^{(t+1)} = \tilde{\Delta}(\tilde{x}_i^{(t)}; \tilde{x}_{-i}^{(t)}, \kappa_i^{(t)}), \qquad \tilde{x}_i^{(0)} = x_i \qquad (10)$$

Considering $\tilde{\Delta}$ as a layer in a neural network, approximating $T$ rounds can be done by stacking.

In Eq. (10), $\kappa_i^{(t)}$ is set to accumulate costs of approximate responses, $\kappa_i^{(t)} = \kappa_i^{(t-1)} + c(\tilde{x}_i^{(t-1)}, \tilde{x}_i^{(t)})$. One observation is that for 2-norm costs, $\kappa_i^{(t)} = c(\tilde{x}_i^{(0)}, \tilde{x}_i^{(t)})$ (by the triangle inequality; since all points move along a line, equality holds). We can therefore simplify Eq. (9) and replace $c(x_i^{(t-1)}, x'_i) - \kappa_i^{(t-1)}$ with $c(x_i^{(0)}, x'_i)$. For other costs, this gives a lower bound (see Appendix B.1).

## 5 EXPERIMENTS

### 5.1 SYNTHETIC DATA

We begin our empirical evaluation by demonstrating different aspects of learning in our setting using a simple but illustrative synthetic example. Additional results and insights on movement trends, the effects of movement on accuracy, and the importance of looking ahead, can be found in Appendix D.1.

For our experimental setup, we set $\ell = 1$ and sample features $x_i \in \mathbb{R}$ for each class from a corresponding Gaussian $\mathcal{N}(y, 1)$ (classes are balanced). For each node, we uniformly sample 5 neighbors from the same class and 3 from the other, and use uniform weights. This creates a task where both features and the graph are informative about labels, but only partially, and in a complementary manner (i.e., noise is uncorrelated; for $i$ with $y_i = 1$, if $x_i < 0$, it is still more likely that most neighbors have $x_j > 0$, and vice versa). As it is a-priori unclear how to optimally combine these sources, we study the effects of relying on the graph to various degrees by varying a global $\alpha$, i.e., setting $\widetilde{w}_{ii} = (1 - \alpha)$ and $\widetilde{w}_{ij} = \alpha/\deg_i$ for all $i$ and all $j \neq i$. We examine both strategic and non-strategic settings, the latter serving as a benchmark. Since $\ell = 1$, $H = \{h_b\}$ is simply the class of thresholds, hence we can scan all thresholds $b$ and report learning outcomes for all models $h_b \in H$. For non-strategic data, the optimal $h^*$ has $b^* \approx 0$; for strategic data, the optimal $h^*$ can be found using line search. Testing is done on disjoint but similarly sampled held-out features and graph.

**The effects of strategic behavior.** Figure 2 (left) presents the accuracy of the learned $\hat{h}$ for varying $\alpha$ and in different settings. In a non-strategic setting (dashed gray), increasing $\alpha$ helps, but if reliance on the graph becomes exaggerated, performance deteriorates ($\alpha \approx 0.7$ is optimal). Allowing users to respond strategically reverses this result: for $\alpha = 0$ (i.e., no graph), responses lower accuracy by

|  | Cora | CiteSeer | PubMed |
|---|---|---|---|
| Naïve | $52.56_{\pm 0.20}$ | $61.06_{\pm 0.10}$ | $19.02_{\pm 0.01}$ |
| Robust (ours) | $77.51_{\pm 0.38}$ | $75.21_{\pm 0.25}$ | $82.41_{\pm 0.38}$ |
| Non-strategic (benchmark) | $87.95_{\pm 0.08}$ | $77.11_{\pm 0.05}$ | $91.34_{\pm 0.03}$ |

Table 1: Test accuracy of different methods. For all results $T = 3$ and $d = 0.25$.

$\approx 0.26$ points; but as $\alpha$ is increased, the gap *grows*, this becoming more pronounced as test-time response rounds progress (blue lines). Interestingly, performance under strategic behavior is *worst* around the previously-optimal $\alpha \approx 0.75$. This shows how learning in a strategic environment—but *neglecting* to account for strategic behavior—can be detrimental. By accounting for user behavior, our approach (orange line) not only recovers performance, but slightly improves upon the non-strategic setting (this can occur when positive points are properly incentivized; see Appendix D.1).

**Sensitivity analysis.** Figure 2 (right) plots the accuracy of all threshold models $h_b$ for increasing values of $\alpha$. For each $\alpha$, performance exhibits a 'bell-curve' shape, with its peak at the optimal $h^*$. As $\alpha$ increases, bell-curves change in two ways. First, their centers *shift*, decreasing from positive values towards zero (which is optimal for non-strategic data); since using the graph limits users' effective radius of movement, the optimal decision boundary can be less 'stringent'. Second, and interestingly, bell-curves become *narrower*. We interpret this as a measure of *tolerance*: the wider the curve, the lower the loss in accuracy when the learned $\hat{h}$ is close to (but does not equal) $h^*$. The figure shows for a subset of $\alpha$-s 'tolerance bands': intervals around $b^*$ that include thresholds $b$ for which the accuracy of $h_b$ is at least $90\%, 95\%$, and $97.5\%$ of the optimum (horizontal lines). Results indicate that larger $\alpha$-s provide *less* tolerance. If variation in $\hat{h}$ can be attributed to the number of examples, this can be interpreted as hinting that larger $\alpha$-s may entail larger sample complexity.

**Number of layers ($T$).** Figure 2 (right) also shows for each bell-curve the accuracy achieved by learned models $\hat{h}$ of increasing depths, $T = 1, \ldots, 4$ (colored dots). For $\alpha = 0$ (no graph), there are no inter-user dependencies, and dynamics converge after one round. Hence, $T = 1$ suffices and is optimal, and additional layers are redundant. However, as $\alpha$ increases, more users move in later rounds, and learning with insufficiently large $T$ results in deteriorated performance. This becomes especially distinct for large $\alpha$: e.g., for $\alpha = 0.9$, performance drops by $\sim 11\%$ when using $T = 1$ instead of the optimal $T = 4$. Interestingly, lower $T$ always result in lower, more 'lenient' thresholds; as a result, performance deteriorates, and more quickly for larger, more sensitive $\alpha$. Thus, the relations between $\alpha$ and $T$ suggest that greater reliance on the graph requires more depth.

## 5.2 EXPERIMENTS ON REAL DATA

**Data.** We use three benchmark datasets used extensively in the GNN literature: Cora, CiteSeer, and PubMed (Sen et al., 2008; Kipf & Welling, 2017), and adapt them to our setting. We use the standard (transductive) train-test split of Sen et al. (2008); the data is made inductive by removing all test-set nodes that can be influenced by train-set nodes (Hamilton et al., 2017). All three datasets describe citation networks, with papers as nodes and citations as edges. Although these are directed relations by nature, the available data include only undirected edges; hence, we direct edges towards lower-degree nodes, so that movement of higher-degree nodes is more influential. As our setup requires binary labels, we follow standard practice and merge classes, aiming for balanced binary classes that sustain strategic movement. Appendix C includes further details. see Appendix D.2 for additional results on strategic improvement, extending neighborhood size, and node centrality and influence.

**Methods.** We compare our robust learning approach to a naïve approach that does not account for strategic behavior (i.e., falsely assumes that users do not move). As a benchmark we report the performance of the naïve model on non-strategic data (for which it is appropriate). All methods are based on the SGC architecture (Wu et al., 2019) as it is expressive enough to effectively utilize the graph, but simple enough to permit rational user responses (Eq. (4); see also notes Sec. 1.1). We use the standard weights $\widetilde{W} = D^{-\frac{1}{2}} A D^{-\frac{1}{2}}$ where $A$ is the adjacency matrix and $D$ is the diagonal degree matrix.

**Optimization and setup.** We train using Adam and set hyperparameters according to Wu et al. (2019) (learning rate=0.2, weight decay=$1.3 \cdot 10^{-5}$). Training is stopped after 20 epochs (this usually suffices

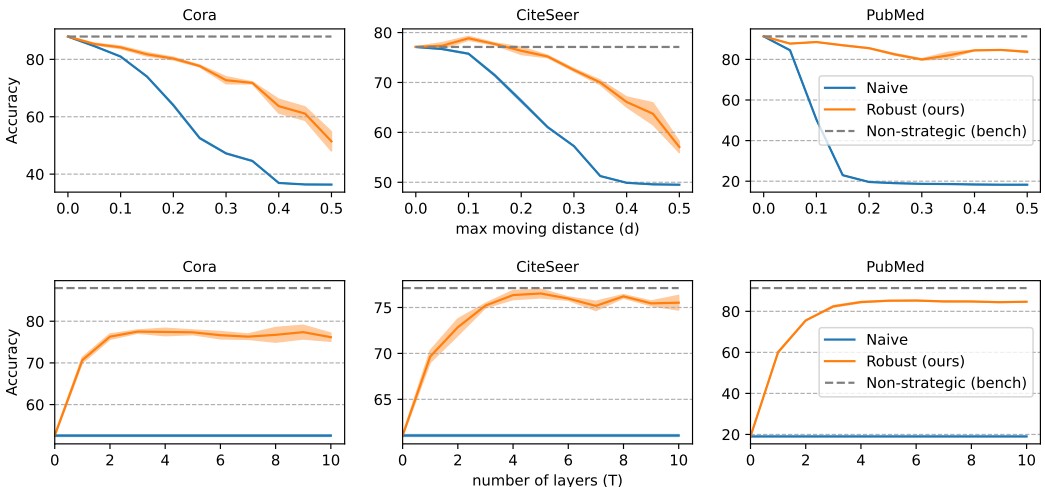

Figure 3: Accuracy for increasing: **(Top:)** max distance $d$ ($T = 3$); **(Bottom:)** depth $T$ ($d = 0.25$).

for convergence). Hyperparameters were determined based only on the train set: $\tau = 0.05$, chosen to be the smallest value which retained stable training, and $T = 3$, as training typically saturates then (we also explore varying depths). We use $\beta$-scaled 2-norm costs, $c_\beta(x, x') = \beta\|x - x'\|_2, \beta \in \mathbb{R}_+$, which induce a maximal moving distance of $d_\beta = 2/\beta$. We observed that values around $d = 0.5$ permit almost arbitrary movement; we therefore experiment in the range $d \in [0, 0.5]$, but focus primarily on the mid-point $d = 0.25$ (note $d = 0$ implies no movement). Mean and standard errors are reported over five random initializations. Appendix C includes further details.

**Results.** Table 1 presents detailed results for $d = 0.25$ and $T = 3$. As can be seen, the naive approach is highly vulnerable to strategic behavior. In contrast, by anticipating how users collectively respond, our robust approach is able to recover most of the drop in accuracy (i.e., from 'benchmark' to 'naïve'; Cora: 35%, CiteSeer: 16%, PubMed: 72%). Note this is achieved with a $T$ much smaller than necessary for response dynamics to converge ($T_{\max}$: Cora=7, CiteSeer=7, PubMed=11).

Fig. 3 (top) shows results for varying max distances $d \in [0, 0.5]$ and fixing $T = 3$ (note $d = 0$ entails no movement). For Cora and CiteSeer, larger max distances—the result of lower modification costs—hurt performance; nonetheless, our robust approach maintains a fairly stable recovery rate over all values of $d$. For PubMed, our approach retains $\approx 92\%$ of the *optimum*, showing resilience to reduced costs. Interestingly, for CiteSeer, in the range $d \in [0.05, 0.15]$, our approach *improves* over the baseline, suggesting it utilizes strategic movements for improved accuracy (as in Sec. 5.1).

Fig. 3 (bottom) shows results for varying depths $T \in \{0, \ldots, 10\}$. For all datasets, results improve as $T$ increases, but saturate quickly at $T \approx 3$; this suggests a form of robustness of our approach to overshooting in choosing $T$ (which due to smoothing can cause larger deviations from the true dynamics). Using $T = 1$ recovers between $65\% - 91\%$ (across datasets) of the optimal accuracy. This shows that while considering only one round of user responses (in which there are no dependencies) is helpful, it is much more effective to consider multiple, dependent rounds—even if only a few.

## 6 DISCUSSION

In this paper we study strategic classification under graph neural networks. Relying on a graph for prediction introduces dependencies in user responses, which can result in complex correlated behavior. The incentives of the system and its users are not aligned, but also not discordant; our proposed learning approach utilizes this degree of freedom to learn strategically-robust classifiers. Strategic classification assumes rational user behavior; this necessitates classifiers that are simple enough to permit tractable best-responses. A natural future direction is to consider more elaborate predictive architectures coupled with appropriate boundedly-rational user models, in hopes of shedding further light on questions regarding the benefits and risks of transparency and model explainability.

ETHICS AND SOCIETAL IMPLICATIONS

In our current era, machine learning is routinely used to make predictions about humans. These, in turn, are often used to inform—or even determine—consequential decisions. That humans can (and do) respond to decision rules is a factual reality, and is a topic of continual interest in fields such as economics (e.g., Nielsen et al., 2010) and policy-making (e.g., Camacho & Conover, 2013); the novelty of strategic classification is that it studies decision rules that are a product of learned predictive models. Strategic classification not only acknowledges this reality, but also proposes tools for learning in ways that account for it. But in modeling and anticipating how users respond, and by adjusting learning to accommodate for their effects—learning also serves to 'steer' its population of users, perhaps inadvertently, towards certain outcomes (Hardt et al., 2022).

GNNs are no exception to this reality. In the domain of graph-based learning, the role of predictive models is expressed in how they associate social connections with decision outcomes for individuals; Clearly, the choice of whether to make use of social data for decisions can be highly sensitive, and doing so necessitates much forethought and care. But the question of whether to use social data to enhance prediction is not a binary in nature, i.e., there is no simple 'right' or 'wrong'. Consider our example of the credit scoring company, Lenddo. On the one hand, Lenddo has been criticized that it may be discriminating applicants based on who they chose to socialize with (or, rather, who chooses to socialize with them). But on the other hand, Lenddo, who focuses primarily on developing countries, has been acclaimed for providing financial assistance to a large community of deserving applicants who, due to conservative norms in typically credit scoring rules, would otherwise be denied a consequential loan.[9]

Such considerations apply broadly. In other focal domains in strategic classification, such as loans, university admissions, and job hiring, the use of social data for informing decisions can be highly controversial, on both ethical and legal grounds. Regulation is necessary, but as in similar areas, often lags far behind the technology itself. This highlights the need for transparency and accountability in how, when, and to what purpose, social data is used (Ghalme et al., 2021; Jagadeesan et al., 2021; Bechavod et al., 2022; Barsotti et al., 2022b).

ACKNOWLEDGEMENTS

This research was supported by the Israel Science Foundation (grant No. 278/22).

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

# A ANALYSIS

## A.1 HITCHHIKING

Here we provide a concrete example of hitchhiking, following Fig. 1 (E). The example includes three nodes, $i, j, k$, positioned at:

$$x_k = -3, \qquad x_i = -2.1 \qquad x_j = -0.5,$$

and connected via edges $k{\to}j$, and $j{\to}i$. Edge weights $\widetilde{w}_{ji} = 0.6$ and $\widetilde{w}_{ii} = 0.4$; $\widetilde{w}_{kj} = 1/3$ and $\widetilde{w}_{jj} = 2/3$; and $\widetilde{w}_{kk} = 1$. The example considers a threshold classifier $h_b$ with $b = 0$, and unit-scale costs (i.e., $\beta = 1$) inducing a maximal moving distance of $d = 2$.

We show that $i$ cannot invest effort to cross and obtain $\hat{y}_i = 1$; but once $j$ moves (to obtain $\hat{y}_j = 1$), this results in $i$ also being classified positively (*without* moving). Initially (at round $t = 0$), node embeddings are:

$$\phi_k = -3 \qquad , \phi_i = -1.14 \qquad , \phi_j = -\frac{4}{3}$$

and all points are classified negatively, $\hat{y}_k = \hat{y}_i = \hat{y}_j = -1$. Notice that $i$ cannot cross the decision boundary even if she moves the maximal cost-feasible distance of $d = 2$:

$$\phi(x_i^{(0)} + 2; x_{i-}^{(0)}) = \widetilde{w}_{ii}(x_i^{(0)} + 2) + \widetilde{w}_{ji}x_j^{(0)} = 0.4(-2.1 + 2) + 0.6(-\frac{1}{2}) = -0.34 < 0$$

Hence, $i$ doesn't move, so $x_i^{(1)} = x_i^{(0)}$. Similarly, $k$ cannot cross, so $x_k^{(1)} = x_k^{(0)}$. However, $j$ *can* cross by moving to 1.5 (at cost 2) in order to get $\hat{y}_j = 1$:

$$x_j^{(1)} = 1.5 = -1/2 + 2 = x_j^{(0)} + 2$$

$$\Rightarrow \phi(x_j^{(1)}; x_{j-}^{(1)}) = \widetilde{w}_{jj}x_j^{(1)} + \widetilde{w}_{kj}x_k^{(0)} = \frac{2}{3}x_j^{(1)} + \frac{1}{3}(-3) = 0 \;\Rightarrow\; \hat{y}_j^{(1)} = 1$$

After $j$ moves, $i$ is classified positively (and so does not need to move):

$$\phi(x_i^{(1)}; x_{i-}^{(1)}) = \widetilde{w}_{ii}x_i^{(1)} + \widetilde{w}_{ji}x_j^{(1)} = 0.4(-2.1) + 0.6\frac{3}{2} = 0.06 > 0 \;\Rightarrow\; \hat{y}_i^{(2)} = 1$$

## A.2 CASCADING BEHAVIOR

We give a constructive example (for any $n$) which will be used to prove Propositions 3 and 4. The construction is modular, meaning that we build a small 'cyclic' structure of size 3, such that for any given $n$, we simply replicate this structure roughly $n/3$ times, and include two additional 'start' and 'finish' nodes. Our example assumes a threshold classifier $h_b$ with $b = 0$, and scale costs $c_\beta$ with $\beta = 1.5$ inducing a maximum moving distance of $d_\beta = 3$.

Let $n$. We construct a graph of size $n + 2$ as follows. Nodes are indexed $0, \ldots, n + 1$. The graph has bi-directional edges between each pair of consecutive nodes, namely $(i, i + 1)$ and $(i + 1, i)$ for all $i = 0, \ldots, n$, except for the last node, which has only an outgoing edge $(n + 1, n)$, but no incoming edge. We set uniform normalized edge weights, i.e., $w_{ij} = 1/3$ and $w_{ii} = 1/3$ for all $1 \le i, j \le n$, and $w_{0,0} = w_{0,1} = 1/2$ and $w_{n+1,n+1} = w_{n+1,n} = 1/2$. The initial features of each node are defined as:

$$x_0 = -1, \qquad x_i = \begin{cases} 2 & \text{if } i \bmod 3 = 1 \\ -4 & \text{o.w.} \end{cases} \qquad \forall i = 1, \ldots, n + 1 \qquad (11)$$

Figure 4 (A) illustrates this for $n = 3$. Note that while the graph creates a 'chain' structure, the positioning of node features is cyclic (starting from $n = 1$): $2, -4, -4, 2, -4, -4, 2, \ldots$ etc.

We begin with a lemma showing that in our construction, each node $i = 1, \ldots, n$ moves precisely at round $t = i$.

**Lemma 1.** *At every round $1 \le t \le n$:*

*(1) node $i = t$ moves, with $x_i^{(t)} = 5$ if $k \bmod 3 = 1$, and $x_i^{(t)} = -1$ otherwise*
*(2) all nodes $j > t$ do not move, i.e., $x_j^{(t)} = x_j^{(t-1)}$*

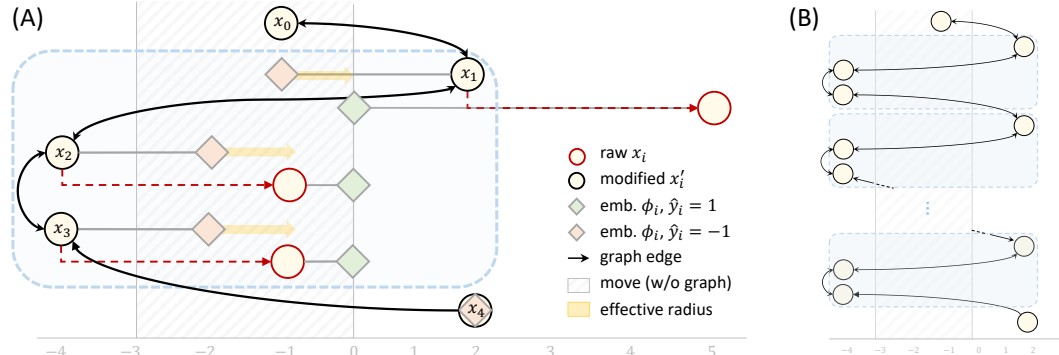

Figure 4: **Cascading behavior**. The construction for Propositions 3 and 4, for: **(A)** $n = 3$, which includes one copy of the main 3-node module (light blue box), and **(B)** for any $n > 3$, by replicating the module in sequence (note $|V| = n + 2$).

Note that (1) (together with Prop. 1) implies that for any round $t$, all nodes $i < t$ (which have already moved at the earlier round $t' = i$) do not move again. Additionally, (2) implies that all $j > t$ remain in their initial position, i.e., $x_j^{(t)} = x_j^{(0)}$. Finally, notice that the starting node $x_0$ has $\phi_0 = 0.5$, meaning that $\hat{y}_0^{(0)} = 1$, and so does not move at any round.

*Proof.* We begin with the case for $n = 3$.

- **Round 1:** Node $i = 1$ can cross by moving the maximal distance of 3:

$$\widetilde{w}_{1,1}(x_1^{(0)} + 3) + \widetilde{w}_{0,1}x_0^{(0)} + \widetilde{w}_{2,1}x_2^{(0)} = \frac{1}{3}(2 + 3) + \frac{1}{3}(-1) + \frac{1}{3}(-4) = 0 \quad (12)$$

However, nodes 2,3 cannot cross even if they move the maximal feasible distance:

$$\widetilde{w}_{2,2}(x_2^{(0)} + 3) + \widetilde{w}_{1,2}x_1^{(0)} + \widetilde{w}_{3,2}x_3^{(0)} = \frac{1}{3}(-4 + 3) + \frac{1}{3}(2) + \frac{1}{3}(-4) = -1 < 0 \quad (13)$$

$$\widetilde{w}_{3,3}(x_3^{(0)} + 3) + \widetilde{w}_{2,3}x_2^{(0)} + \widetilde{w}_{4,3}x_4^{(0)} = \frac{1}{3}(-4 + 3) + \frac{1}{3}(-4) + \frac{1}{3}(2) = -1 < 0 \quad (14)$$

- **Round 2:** Node $i = 2$ can cross by moving the maximal distance of 3:

$$\widetilde{w}_{2,2}(x_2^{(1)} + 3) + \widetilde{w}_{1,2}x_1^{(1)} + \widetilde{w}_{3,2}x_3^{(1)} = \frac{1}{3}(-4 + 3) + \frac{1}{3}(5) + \frac{1}{3}(-4) = 0 \quad (15)$$

However, node 3 cannot cross even if it moves the maximal feasible distance:

$$\widetilde{w}_{3,3}(x_3^{(1)} + 3) + \widetilde{w}_{2,3}x_2^{(1)} + \widetilde{w}_{4,3}x_4^{(1)} = \frac{1}{3}(-4 + 3) + \frac{1}{3}(-4) + \frac{1}{3}(2) = -1 < 0 \quad (16)$$

- **Round 3:** Node $i = 3$ can cross by moving the maximal distance of 3:

$$\widetilde{w}_{3,3}(x_3^{(2)} + 3) + \widetilde{w}_{2,3}x_2^{(2)} + \widetilde{w}_{4,3}x_4^{(2)} = \frac{1}{3}(-4 + 3) + \frac{1}{3}(-1) + \frac{1}{3}(2) = 0 \quad (17)$$

Fig. 4 (A) illustrates this procedure for $n = 3$.

Next, consider $n > 3$. Due to the cyclical nature of feature positioning and the chain structure of our graph, we can consider what happens when we sequentially add nodes to the graph. By induction, we can show that:

- $n \mod 3 = 1$: Consider round $t = n$. Node $n$ has $x_n^{(t-1)} = 2$, and two neighbors: $n - 1$, who after moving at the previous round has $x_{n-1}^{(t-1)} = -1$; and $n + 1$, who has a fixed $x_{n+1}^{(t-1)} = -4$. Thus, it is in the same configuration as node $i = 1$, and so its movement follows Eq. (12).

- $n \mod 3 = 2$: Consider round $t = n$. Node $n$ has $x_n^{(t-1)} = -4$, and two neighbors: $n - 1$, who after moving at the previous round has $x_{n-1}^{(t-1)} = 5$; and $n + 1$, who has a fixed $x_{n+1}^{(t-1)} = -4$. Thus, it is in the same configuration as node $i = 2$, and so its movement follows Eq. (15).

- $n \bmod 3 = 0$: Consider round $t = n$. Node $n$ has $x_n^{(t-1)} = -4$, and two neighbors: $n - 1$, who after moving at the previous round has $x_{n-1}^{(t-1)} = -1$; and $n + 1$, who has a fixed $x_{n+1}^{(t-1)} = 2$. Thus, it is in the same configuration as node $i = 2$, and so its movement follows Eq. (17).

Fig. 4 (B) illustrates this idea for $n > 3$.

$\square$

We now proceed to prove the propositions.

**Proposition 3:** The proposition follows immediately from Lemma 1; the only detail that remains to be shown is that node $n + 1$ does not move at all. To see this, note that since it does not have any incoming edges, its embedding depends only on its own features, $x_{n+1}$. If $n + 1 \bmod 3 = 1$, we have $x_{n+1} = 2$, and so $\hat{y}_{n+1} = 1$ without movement. Otherwise, $x_{n+1} = -4$, meaning that it is too far to cross.

**Proposition 4:** Fix $n$ and $k \leq n$. Consider the same construction presented above for a graph of size $k + 2$ Then, add $n - k$ nodes identical nodes: for each $k < j \leq n$, add an edge $k \rightarrow j$, and set $x_j = x_k - 6$. We claim that all such nodes will move exactly at round $k$. Consider some node $k < j \leq n$. Since $x_k$ moves only at round $k$ (following Lemma 1), $j$ does not move in any of the first $t \leq k$ rounds:

$$\widetilde{w}_{j,j}(x_j^{(0)} + 3) + \widetilde{w}_{k,j}x_k^{(0)} = \frac{1}{2}(-x_k^{(0)} - 6 + 3) + \frac{1}{2}(x_k^{(0)}) = \frac{1}{2}(-x_k^{(0)} - 3) + \frac{1}{2}(x_k^{(0)}) = -1.5 < 0 \quad (18)$$

At the end of round $t = k$, node $k$ has a value of $x_k^{(0)} + 3$. This enables $j$ to cross by moving the maximal distance of 3:

$$\widetilde{w}_{j,j}(x_j^{(k)} + 3) + \widetilde{w}_{k,j}x_k^{(k)} = \frac{1}{2}(-x_k^{(0)} - 6 + 3) + \frac{1}{2}(x_k^{(k)}) = \frac{1}{2}(-x_k^{(0)} - 3) + \frac{1}{2}(x_k^{(0)} + 3) = 0 \quad (19)$$

As this applies to all such $j$, we get that $n - k$ nodes move at round $k$, which concludes our proof.

Note the graph is such that, for $b = 0$, without strategic behavior the graph is useful for prediction (increases accuracy from 66% to 100%), so that a learner that is unaware of (or does not account for) strategic behavior is incentivized to utilize the graph. However, once strategic behavior is introduced, naïvely using the graph causes performance to drop to 0%.

## B  OPTIMIZATION

### B.1  PROJECTION

We prove for 2-norm-squared costs. Correctness holds for 2-norm-costs since the argmin is the same (squared over positives is monotone). Calculation of $x_i$'s best response requires solving the following equation:

$$\min_{x'} c(x_i', x_i) \quad \text{s.t} \quad \theta^\top \phi(x_i'; x_{-i}) + b = 0$$

$$\min_{x'} \|x_i' - x_i\|_2^2 \quad \text{s.t} \quad \theta^\top \phi(x_i'; x_{-i}) + b = 0$$

To solve for $x'$, we apply the Lagrange method. Define the Lagrangian as follows:

$$L(x_i', \lambda) = \|x_i' - x_i\|_2^2 + \lambda[\theta^\top \phi(x_i'; x_{-i}) + b]$$

Next, to find the minimum of $L$, derive with respect to $x_i'$, and compare to 0:

$$2(x_i' - x_i) + \lambda\theta\widetilde{w}_{ii} = 0$$

$$x_i' = x_i - \frac{\lambda\widetilde{w}_{ii}}{2}\theta$$

Plugging $x_i'$ into the constraint gives:

$$\theta^\top[\widetilde{w}_{ii}(x_i - \frac{\lambda\widetilde{w}_{ii}}{2}\theta) + \sum_{j\neq i}\widetilde{w}_{ij}x_j] + b = 0$$

$$\theta^\top[\phi(x_i; x_{-i}) - \frac{\lambda\widetilde{w}_{ii}^2}{2}\theta] + b = 0$$

$$\theta^\top\phi(x_i; x_{-i}) + b = \frac{\lambda\widetilde{w}_{ii}^2}{2}\|\theta\|_2^2$$

$$2\frac{\theta^\top\phi(x_i; x_{-i}) + b}{\|\theta\|_2^2\widetilde{w}_{ii}^2} = \lambda$$

Finally, plugging $\lambda$ into the expression for $x_i'$ obtains:

$$x_i' = x_i - \frac{\theta^\top\phi(x_i; x_{-i}) + b}{\|\theta\|_2^2\widetilde{w}_{ii}}\theta$$

### B.2 Generalized Costs

Here we provide a formula for computing projections in closed form for generalized quadratic costs:

$$c(x, x') = \frac{1}{2}(x' - x)^\top A(x' - x)$$

for positive-definite $A$. As before, the same formula holds for generalized 2-norm costs (since the argmin is the same). Begin with:

$$\min_{x'} c(x_i', x_i) \quad \text{s.t} \quad \theta^\top\phi(x_i'; x_{-i}) + b = 0$$

$$\min_{x'} \frac{1}{2}(x_i' - x_i)^\top A(x_i' - x_i) \quad \text{s.t} \quad \theta^\top\phi(x_i'; x_{-i}) + b = 0$$

As before, apply the Lagrangian method:

$$\frac{1}{2}(x_i' - x_i)^\top A(x_i' - x_i) + \lambda[\theta^\top\phi(x_i'; x_{-i}) + b]$$

Derivation w.r.t. to $x_i'$:

$$\frac{1}{2}[A^\top(x_i' - x_i) + A(x' - x)] + \lambda\theta\widetilde{w}_{ii} = 0$$

$$(A^\top + A)x_i' = (A^\top + A)x_i - 2\lambda\theta\widetilde{w}_{ii}$$

Since the matrix $(A^\top + A)$ is PD, we can invert to get:

$$x_i' = x_i - 2\lambda(A^\top + A)^{-1}\theta\widetilde{w}_{ii}$$

Plugging $x_i'$ in the constrain:

$$\theta^\top[\widetilde{w}_{ii}(x_i - 2\lambda(A^\top + A)^{-1}\theta\widetilde{w}_{ii}) + \sum_{j\neq i}\widetilde{w}_{ij}x_j] + b = 0$$

$$\theta^\top[\phi(x_i; x_{-i}) - 2\lambda(A^\top + A)^{-1}\widetilde{w}_{ii}^2\theta] + b = 0$$

$$\theta^\top\phi(x_i; x_{-i}) + b = 2\lambda\theta^\top(A^\top + A)^{-1}\theta\widetilde{w}_{ii}^2$$

Since $(A^\top + A)^{-1}$ is also PD, we get $\theta^\top(A^\top + A)^{-1}\theta > 0$, and hence:

$$\frac{\theta^\top\phi(x_i; x_{-i}) + b}{2\theta^\top(A^\top + A)^{-1}\theta\widetilde{w}_{ii}^2} = \lambda$$

Finally, pluging in $\lambda$:

$$x_i' = x_i - \frac{\theta^\top\phi(x_i; x_{-i}) + b}{\theta^\top(A^\top + A)^{-1}\theta\widetilde{w}_{ii}^2}(A^\top + A)^{-1}\theta\widetilde{w}_{ii}$$

$$x_i' = x_i - \frac{\theta^\top\phi(x_i; x_{-i}) + b}{\theta^\top(A^\top + A)^{-1}\theta\widetilde{w}_{ii}}(A^\top + A)^{-1}\theta$$

Setting $A = I$ recovers Eq. (7).

Table 2: Real data – statistics and experimental details

| | $|V|$ | $|E|$ | $\ell$ | $n_{\text{train}}$ | $n_{\text{test}}^*$ | negative classes | positive classes | $\%\{y = 1\}$ (train / test) |
|---|---|---|---|---|---|---|---|---|
| Cora | 2,708 | 5,728 | 1,433 | 640 | 577 | 0,2,3 | 1,4,5,6 | 44% / 36% |
| CiteSeer | 3,327 | 4,552 | 3,703 | 620 | 721 | 0,2,3 | 1,4,5 | 50% / 49% |
| Pubmed | 19,717 | 44,324 | 500 | 560 | 941 | 1,2 | 0 | 21% / 18% |

### B.3 Improving numerical stability by adding a tolerance term

Theoretically, strategic responses move points precisely on the decision boundary. For numerical stability in classifying (e.g., at test time), we add a small tolerance term, `tol`, that ensures that points are projected to lie strictly within the positive halfspace. Tolerance is added as follows:

$$\min_{x'} c(x_i', x_i) \quad \text{s.t} \quad \theta^\top \phi(x_i; x_{-i}) + b \geq \texttt{tol} \tag{20}$$

This necessitates the following adjustment to Eq. (7):

$$\text{proj}_h(x_i; x_{-i}) = x_i - \frac{\theta^\top \phi(x_i; x_{-i}) + b - \texttt{tol}}{\|\theta\|_2^2 \widetilde{w}_{ii}} \theta \tag{21}$$

However, blindly applying the above to Eq. (8) via:

$$\text{proj}_h^+(x_i; x_{-i}) = x_i - \min\left\{0, \frac{\theta^\top \phi(x_i; x_{-i}) + b - \texttt{tol}}{\|\theta\|_2^2 \widetilde{w}_{ii}}\right\} \theta \tag{22}$$

is erroneous, since any user whose score is lower than `tol` will move—although in principal she shouldn't. To correct for this, we adjust Eq. (8) by adding a mask that ensures that only points in the negative halfspace are projected:

$$\text{proj}_h(x_i; x_{-i}) = x_i - \mathbb{1}\{\theta^\top \phi(x_i; x_{-i}) + b < 0\} \cdot \left(\frac{\theta^\top \phi(x_i; x_{-i}) + b - \texttt{tol}}{\|\theta\|_2^2 \widetilde{w}_{ii}} \theta\right) \tag{23}$$

## C Additional experimental details

**Data.** We experiment with three citation network datasets: Cora, CiteSeer, and Pubmed Sen et al. (2008). Table 2 provides summary statistics of the datasets, as well as experimental details.

**Splits.** All three datasets include a standard train-validation-test split, which we adopt for our use.[10] For our purposes, we use make no distinction between 'train' and 'validation', and use both sets for training purposes. To ensure the data is appropriate for the inductive setting, we remove from the test set all nodes which can be influenced by train-set nodes—this ranges from 6%-43% of the test set, depending on dataset (and possibly setting; see Sec. D.2.1). In Table 2, the number of train samples is denoted $n_{\text{train}}$, and the number of inductive test samples is denoted $n_{\text{test}}^*$ (all original transductive test sets include 1,000 samples).

**Binarization.** To make the data binary (original labels are multiclass), we enumerated over possible partitions of classes into 'negative' and 'positive', and chose the most balanced partition. Experimenting with other but similarly-balanced partitions resulted in similar performance (albeit at times less distinct strategic movement). The exception to this was PubMed (having only three classes), for which the most balanced partition was neither 'balanced' nor stable, and so here we opted for the more stable alternative. Reported partitions and corresponding negative-positive ratios (for train and for test) are given in Table 2.

**Strategic responses.** At test time, strategic user responses are computed by simulating the response dynamics in Sec. 3.1 until convergence.

---

[10]Note that nodes in these sets do not necessarily account for all nodes in the graph.

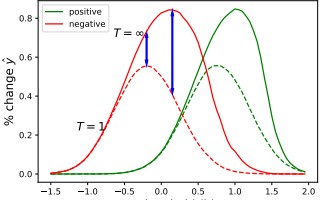 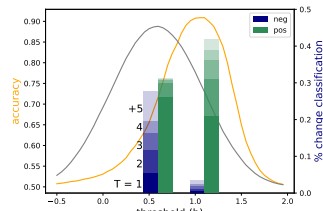 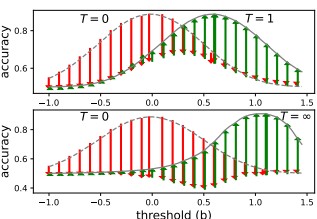

Figure 5: **Synthetic data**: relations between movement and accuracy. All results use $\alpha = 0.7$. **(Left:)** The relative number of points that move for every threshold $b$, per class, and comparing one round ($T = 1$) to convergence ($T = \infty$). **(Center:)** Accuracy for every threshold $b$, after one round ($T = 1$) and at convergence ($T = \infty$). For each optimal $b$, bars show the relative number of points (per class) that obtain $\hat{y} = 1$ due to strategic behavior. **(Right:)** Accuracy for $T = 1$ (top) and $T = \infty$ (bottom), relative to the non-strategic benchmark ($T = 0$), and as a result of strategic movement of negative points (red arrows; decrease accuracy) and positive points (green arrows; improve accuracy).

# D ADDITIONAL EXPERIMENTAL RESULTS

## D.1 EXPERIMENTS ON SYNTHETIC DATA

In this section we explore further in depth the relation between user movement and classification performance, using our synthetic setup in Sec. 5.1 (all examples discussed herein use $\alpha = 0.7$). From a predictive point of view, graphs are generally helpful if same-class nodes are well-connected. This is indeed the case in our construction (as can be seen by the performance of the benchmark method with non-extreme $\alpha > 0$ values). From a strategic perspective, however, connectivity increases cooperation, since neighboring nodes can positively influence each other over time. In our construction, cooperation occurs mostly within classes, i.e., negative points that move encourage other negative points to move, and similarly for positive points.

**Movement trends.** Fig. 5 (left) shows how different threshold classifiers $h_b$ induce different degrees of movements. The plot shows the relative number of points (in percentage points) whose predictions changed as a result of strategic behavior, per class (red: $y = -1$, green: $y = 1$) and over time: after one round ($T = 1$, dashed lines), and at convergence ($T = \infty$, solid lines). As can be seen, there is a general trend: when $b$ is small, mostly negative points move, but as $b$ increases, positive points move instead. The interesting point to observe is the gap between the first round ($T = 1$) and final round ($T = \infty$). For negative points, movement at $T = 1$ peaks at $b_1 \approx -0.25$, but triggers relatively little consequent moves. In contrast, the peak for $T = \infty$ occurs at a larger $b_\infty \approx 0.15$. For this threshold, though *less* points move in the first round, these trigger significantly *more* additional moves at later rounds—a result of the connectivity structure within the negative cluster of nodes (blue arrows). A similar effect takes place for positive nodes.

**The importance of looking ahead.** Fig. 5 (center) plots for a range of thresholds $b$ the accuracy of $h_b$ at convergence ($T = \infty$; orange line), and after one round ($T = 1$; gray line). The role of the latter is to illustrate the outcomes as 'perceived' by a myopic predictive model that considers only one round (e.g., includes only one response layer $\tilde{\Delta}$); the differences between the two lines demonstrate the gap between perception (based on which training chooses a classifier $\hat{h}$) and reality (in which the classifier $\hat{h}$ is evaluated). As can be seen, the mypoic approach leads to an under-estimation of the optimal $b^*$; at $b_1 \approx 0.5$, performance for $T = 1$ is optimal, but is severely worse under the true $T = \infty$, for which optimal performance is at $b_\infty \approx 1.15$.

The figure also gives insight as to *why* this happens. For both $b_1$ and $b_\infty$, the figure shows (in bars) the relative number of points from each class who obtain $\hat{y} = 1$ as a result of strategic moves. Bars are stacked, showing the relative number of points that moved per round $T$ (darker = earlier rounds; lightest = convergence). As can be seen, at $b_1$, the myopic models believes that many positive points, but only few negative points, will cross. However, in reality, at convergence, the number of positive points that crossed is only slightly higher than that of negative points. Hence, the reason for the(erroneous) optimism of the myopic model is that it did not correctly account for the magnitude of

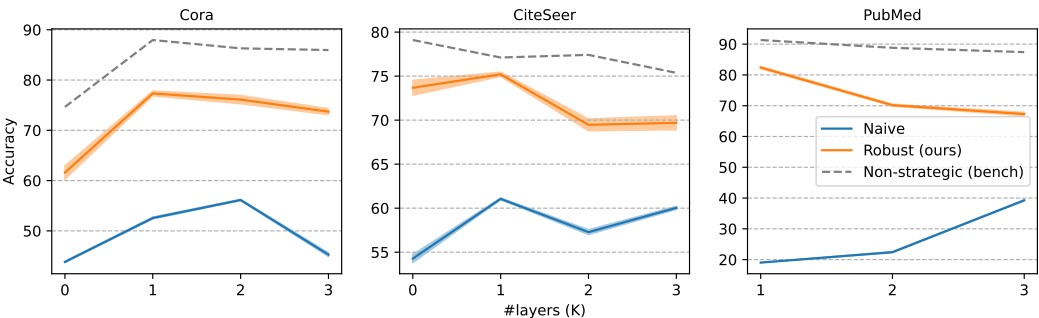

Figure 6: Accuracy for varying number of SGC layers ($K$). Note test sets may vary across $K$.

correlated moves of negative points, which is expressed over time. In contrast, note that at $b_\infty$, barely any negative points cross.

**How movement affects accuracy.** An important observation about the relation between movement and accuracy is that for any classifier $h$, any *negative* point that moves *hurts* accuracy (since $y = -1$ but predictions become $\hat{y} = 1$), whereas any *positive* point that moves *helps* accuracy (since $y = 1$ and predictions are now $\hat{y} = 1$). Fig. 5 (right) shows how these movements combine to affect accuracy. The figure compares accuracy before strategic behavior ($T = 0$; dashed line) to after one response round ($T = 1$; solid line, top plot) and to convergence ($T = \infty$; solid line, lower plot). As can be seen, for any $b$, the difference between pre-strategic and post-strategic accuracy amounts to exactly the degradation due to negative points (red arrows) plus the improvement of positive points (green arrows).

Note, however, the difference between $T = 1$ and $T = \infty$, as they relate to the benchmark model ($T = 0$, i.e., no strategic behavior). For $T = 1$ (top), across the range of $b$, positive and negative moves roughly balance out. A result of this is that curves for $T = 0$ and $T = 1$ are very much similar, and share similar peaks in terms of accuracy (both have $\approx 0.89$). One interpretation of this is that if points were permitted to move only one round, the optimal classifier can completely recover the benchmark accuracy by ensuring that the number of positive points the moves overcomes the number of negative points. However, for $T = \infty$ (bottom), there is a *skew* in favor of positive points (green arrows). The result of this is that for the optimal $b$, additional rounds allow positive points to move in a way that obtains slightly *higher* accuracy (0.91) compared to the benchmark (0.89). This is one possible mechanism underlying our results on synthetic data in Sec. 5.1, and later for our results on real data in Sec. 5.2.

### D.2 EXPERIMENTS ON REAL DATA

#### D.2.1 EXTENDING NEIGHBORHOOD SIZE

One hyperparameter of SGC is the number 'propagation' layers, $K$, which effectively determines the graph distance at which nodes can influence others (i.e., the 'neighborhood radius'). Given $K$, the embedding weights are defined as $\widetilde{W} = D^{-\frac{1}{2}} A^K D^{-\frac{1}{2}}$ where $A$ is the adjacency matrix and $D$ is the diagonal degree matrix. For $K = 0$, the graph is unused, which results in a standar linear classifier over node features. Our results in the main body of the paper use $K = 1$.

Fig. 6 shows results for an increasing $K$ (we set $T = 3, d = 0.25$ as in our main results). Results are mixed: for PubMed, higher $K$ seems to lead to less drop in accuracy for naïve and less recovery for our approach; for Cora and CiteSeer, results are unstable. Note however that this may likely be a product of our inductive setup: since varying $K$ also changes the effective test set (since to preserve inductiveness, larger $K$ often necessitates removing more nodes), test sets vary across conditions and decrease in size, making it difficult to directly compare result across different $K$.

### D.2.2 STRATEGIC IMPROVEMENT

Our main results in Sec. 5.2 show that for CiteSeer, our strategically-aware approach outperforms the non-strategic benchmark (similarly to our synthetic experiments). Here we show that these results are robust. Fig. 7 provides higher-resolution results on CiteSeer for max distances $d \in [0, 0.22]$ in hops of $0.01$. All other aspects the setup match the original experiment. As can be seen, our approach slightly but consistently improves upon the benchmark until $d \approx 0.17$.

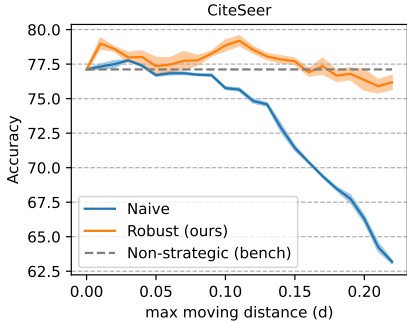

Figure 7: Accuracy on CiteSeer for increasing max distance $d$ (focused and higher-resolution).

### D.3 NODE CENTRALITY AND INFLUENCE

In this experiment we set to explore the role played by central nodes in the graph in propagating the influence of strategic behavior. Since the embedding of a node $i$ is partly determined by its in-neighbors, broadly we would expect that nodes have high out-degree would be highly influential: as 'anchors' that prevent others from moving if they themselves do not, and as 'carriers' which either push neighbors over the boundary—or at least promote the closer to it—if they do move.

**Experimental setup.** To study the role of such nodes, we preform the following experiment. First, we order the nodes by decreasing out-degree, so that the potentially more influential nodes appear first in the ranking. Then, for each $q \in \{0, 10, 20, ..., 100\}$, we disconnect nodes in the $q^{\text{th}}$ percentile, i.e., remove all edges emanating from the top-$q\%$ ranked nodes. For each such condition, we examine learning and its outcomes, and compare performance to a control condition in which nodes are ordered randomly. The difference in performance and other learning outcomes provides us with a measure of the importance of high-degree nodes.

**Results.** Figure 8 shows results for all methods (naïve, robust (ours), and the non-strategic benchmark) and across all three datasets (Cora, CiteSeer, and Pubmed). In all conditions, we vary on the x-axis the portion of nodes that remain connected, where at zero (left end) we get an empty graph, and at 100 (right end) we get the original graph. Note that the y-axis varies in scale across plots.

First, consider Cora. In terms of accuracy (upper plots), results show that on the benchmark (evaluated in a non-strategic environment, in which users do not move), the general trend is that more edges help improve performance. However, the gain in performance is much more pronounced for high-degree nodes. Interestingly, for the naïve method (which operates on strategically modified data, but does not anticipate updates), the trend is reversed: user utilize edges in a way that is detrimental to performance—and more so when high-degree nodes remain, making them a vulnerability. Our robust approach is also sensitive to the addition of edges, but to a much lesser degree (the drop is performance is minor); moreover, which nodes are disconnected appears to make little difference, which we take to mean that our approach can counter the dominant role of central nodes.

The lower plots, which describe the portion of users that moves and the portion of users that crossed, provides some explanation as to how this is achieved. For the naïve approach, nearly half of all user move—and all users that move, also cross. This occurs faster (in terms of the portion of nodes removed) in the degree condition. In our robust approach, note that the number of nodes that move is halved, and of those, not all cross, which demonstrates how are learning objective, which anticiaptes strategic behavior, can act to prevent it.

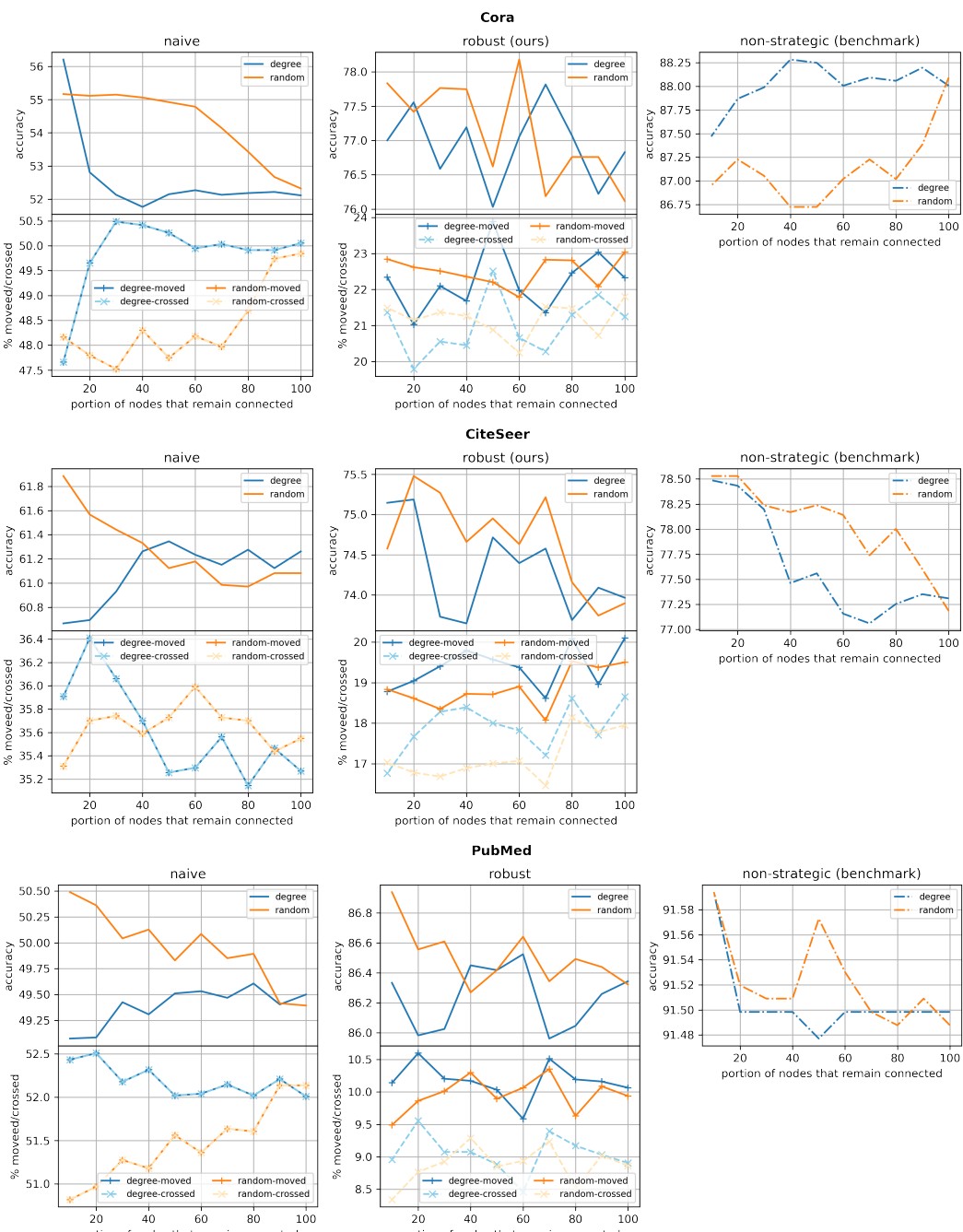

Figure 8: The effect of node centrality on performance and strategic behavior across datasets.

For CiteSeer and PubMed, the general trend in accuracy is reversed for the non-strategic benchmark, and for the naïve approach, begins with a gap, which closes at some point (sooner in CiteSeer). Despite these differences, the qualitative behavior of our robust classifier is similar to Cora, and achieves fairly stable accuracy (with a mild negative slope) in both conditions and for both datasets. As in citeseer, movement and crossing behavior is similar, in that for the naïve approach a considerable portion of users move and cross (with a gap between conditions in PubMed); and in that our robust approach greatly reduces the number of users that move, and even more so the number users that cross.

# E  ADDITIONAL ANALYTIC RESULTS

## E.1  PERFORMANCE GAPS: ROBUST LEARNING VS. THE NON-STRATEGIC BENCHMARK

When learning a robust classifier on strategic data, intuitively we may hope that its performance approaches that of the optimal classifier on non-strategic data, which we may think of as a target upper bound. A natural question to then ask is - can we always reach this upper bound and close the performance gap? Here we answer this question in the negative, by showing a simple example where the optimal classifier on non-strategic data achieves perfect accuracy, but the optimal classifier on strategic data achieves only 0.66 accuracy. We then show that by introducing a mild change—namely slightly shifting the features of one node—the performance gap closes entirely. This, combined with our result from the previous Sec. D.2.2 showing that the gap can also be *negative*, highlights how the gap in performance greatly depends on the structure of the input (i.e., the graph, features, and labels), and in a way which can be highly sensitive even to minor changes.

### E.1.1  LARGE GAP

Our first example in which the best obtainable gap is 0.33 is shown in Figure 9 **(Left)**. The example has three nodes described by one-dimensional features $x_1 = 1.2$, $x_2 = -1$, $x_3 = -1$ and with labels $y_1 = 1$, $y_2 = -1$, $y_3 = -1$. We use the standard cost function so that maximal distance to move is $d_\beta = 2$, and use uniform edge weights. Since $x \in \mathbb{R}$, classifiers are simply threshold functions on the real line, defined by a threshold parameter $b \in \mathbb{R}$.

First, we demonstrate that for non-strategic data, there exists a 'benchmark' classifier that achieves perfect accuracy. Let $b = 0$. Node embeddings are:

$$\phi_1 = \frac{x_1 + x_2}{2} = \frac{1 - 1}{2} = 0 = b$$
$$\phi_2 = \frac{x_1 + x_2 + x_3}{3} = \frac{1 - 1 - 1}{3} = -\frac{1}{3} < 0 = b$$
$$\phi_3 = \frac{x_2 + x_3}{2} = \frac{-1 - 1}{2} = -1 < 0 = b$$

This give predictions $\hat{y}_1 = +1 = y_1, \hat{y}_2 = -1 = y_2, \hat{y}_3 = -1 = y_3$, which are all correct.

Next. we prove that there is no robust classifier capable of achieving perfect accuracy on strategic data. Suppose by contradiction that such a classifier exists, denoted $b$. For $x_2$ to move in the first round, the following conditions must hold:

$$\frac{x_1 + x_2' + x_3}{3} = b, \qquad 1 + x_2' - 1 = 3b, \qquad x_2' = 3b$$

Since $x_2$ moves at most distance 2, it moves in the first round only if $-\frac{1}{3} < b \leq \frac{1}{3}$. However, if it does move - it ends up getting an incorrect prediction. In addition, in this case $x_3$ can also get a positive prediction (either in the first round or in the next round, depending on whether $b > 0$), in which case the accuracy is $\frac{1}{3}$. Thus, we get that $\frac{1}{3} < b$ (note that for $b < -\frac{1}{3}$ $x_2$ is classified as positive which is wrong).

Next, we look at the behavior of $x_1$ in the first round. Conditions for movement are:

$$\frac{x_1' + x_2}{2} = b, \qquad x_1' - 1 = 2b, \qquad x_1' = 2b + 1$$

Here $x_1$ gets negative classification if $b > 0$. If $b > 1$, then $x_1$ does not move, since the distance required is larger than 2. Thus, $x_1$ does move (beyond the classifier) and gets the correct prediction only if $0 < b \leq 1$.

However, considering now the second round of movement for $x_2$ (which only occurs if $b > \frac{1}{3}$ since for $0 < b \leq \frac{1}{3}$ $x_2$ moves at the first round), we get the conditions:

$$\frac{x_1' + x_2' + x_3}{3} = b, \qquad 2b + 1 + x_2' - 1 = 3b, \qquad x_2' = b$$

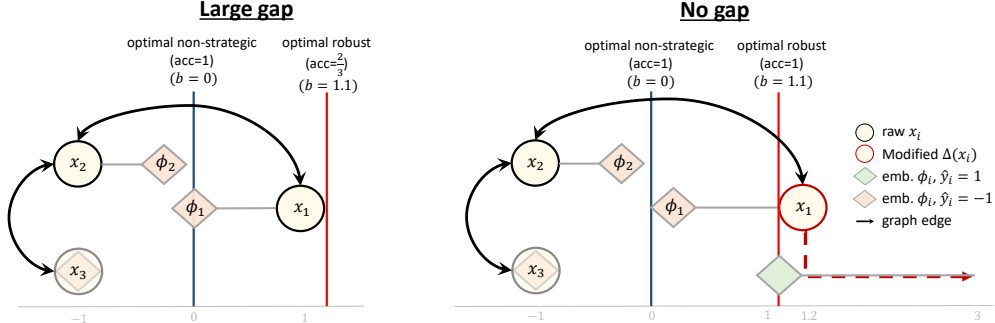

Figure 9: An example of how a small change in features leads to a large decrease in the performance of the robust model (**Left:**) An optimal robust classifier achieves an accuracy of 1. (**Right**) The optimal robust classifier achieves an accuracy of $\frac{2}{3}$ (there is no movement in that case).

This means that for all $0 < b \le 1$, $x_2$ moves and gets an *incorrect* prediction. This occurs either for $x_1$ or for $x_2$. Consequently, there is no $b$ that achieves an accuracy of 1, which contradicts the assumption.

The optimal accuracy of any robust classifier for this example is $\frac{2}{3}$, which is achieved by any $b > 1$. In this case, none of the nodes move, and $x_1$ is classified negatively, whereas its true label is positive.

### E.1.2   NO GAP

We now give a nearly identical example, shown in Figure 9 (**Right**), in which the gap becomes zero.

We use the same example as before, but set $x_1 = 1$ (instead of $x_1 = 1.2$). We begin by showing that there does exist a classifier which achieves perfect accuracy on non-strategic data. Let $b = 0$, then embeddings are now:

$$\phi_1 = \frac{x_1 + x_2}{2} = \frac{1.2 - 1}{2} = 0.1 > 0 = b$$

$$\phi_2 = \frac{x_1 + x_2 + x_3}{3} = \frac{1.2 - 1 - 1}{3} = -\frac{4}{15} < 0 = b$$

$$\phi_3 = \frac{x_2 + x_3}{2} = \frac{-1 - 1}{2} = -1 < 0 = b$$

(note that since all nodes are connected in the graph, changing $x_1$ requires us to recompute all embeddings). Predictions now become:

$$\hat{y}_1 = +1 = y_1, \qquad \hat{y}_2 = -1 = y_2, \qquad \hat{y}_3 = -1 = y_3$$

which are all correct.

Next, we show that there also exists a classifier which achieves perfect accuracy no *strategic* data. Let $b = 1.1$. In the first round, $x_1$ moves, and flips its predicted label:

$$\phi_1 = \frac{x_1' + x_2}{2} = \frac{x_1 + 2 + x_2}{2} = \frac{x_1 + 2 + x_2}{2} = \frac{3.2 - 1}{2} = 1.1 = b$$

Here, even if the other nodes move to the fullest extent, they do not have sufficient influence to revert this prediction:

$$\phi_2 = \frac{x_1 + x_2' + x_3}{3} = \frac{1.2 + -1 + 2 - 1}{3} = 0.4 < 1.1 = b$$

$$\phi_3 = \frac{x_2 + x_3'}{2} = \frac{-1 - 1 + 2}{2} = 0 < 1.1 = b$$

Thus, we get

$$\hat{y}_1 = +1 = y_1, \qquad \hat{y}_2 = -1 = y_2, \qquad \hat{y}_3 = -1 = y_3$$

which are also all correct.

### E.2 STRATEGIC BEHAVIOR OF CLIQUES

Here we give a result that considers graph structure. In particular, we consider cliques, and show that for uniform weights, either all nodes move together—or none do.

**Proposition 6.** *Consider $n$ nodes which are all fully connected, i.e., form a clique, and assume uniform edge weights. Then for any dimension $d$, any assignment of features $x_1, \ldots, x_n \in \mathbb{R}^d$, and for any classifier $h$, either (i) all $n$ nodes move in the first round, or (ii) none of the nodes move at all.*

*Proof.* Consider the case in which at least one node $i$ moves in the first round. Denote by $z$ the change in $x_i$ made in order to cross the classifier, i.e., $z = x_i^{(1)} - x_i^{(0)}$. Note that in order for $x_i$ to move, the following conditions must be satisfied:

$$||z||_2 \leq 2, \qquad \theta^T \phi(x_i^{(1)}, x_{-i}^{(0)}) + b = 0$$

We now show that every other node $t$, if it moves a distance of $z$, then it can also cross the classifier in the first round:

$$
\begin{aligned}
\phi(x_i^{(1)}; x_{-i}^{(0)}) &= \frac{x_i^{(1)} + \sum_{j \neq i} x_j^{(0)}}{n} \\
&= \frac{x_i^{(0)} + z + \sum_{j \neq i} x_j^{(0)}}{n} \\
&= \frac{z + \sum_{j=1}^{n} x_j^{(0)}}{n} \\
&= \frac{x_t^{(0)} + z + \sum_{j \neq t} x_j^{(0)}}{n} \\
&= \frac{x_t^{(1)} + \sum_{j \neq t} x_j^{(0)}}{n} = \phi(x_t^{(1)}; x_{-t}^{(0)})
\end{aligned}
$$

Hence, we get that $\phi(x_i^{(1)}; x_{-i}^{(0)}) = \phi(x_t^{(1)}; x_{-t}^{(0)})$, and thus $\theta^T \phi(x_t^{(1)}, x_{-t}^{(0)}) + b = 0$, which indicates that $x_t$ can also cross the classifier in the first round. $\square$

### E.3 GRAPH DIAMETER AND THE SPREAD OF INFORMATION

Since nodes that move can influence outcomes for their neighbors, intuitively we might expect that the diameter of the graph would be an upper bound on the reach of such influence. Here we give a negative result which shows that influence can far exceed the diameter.

**Proposition 7.** *Let $G$ be a graph with diameter $d$, then the number of update rounds is* not *bounded by $d + c$, for any constant $c$.*

*Proof.* We prove by giving an example in which the number of rounds until convergence, $T$, is roughly *twice* the diameter.

We use the same example as in Sec. A.2, but with the following small changes: (i) we add edge from $x_n$ to $x_{n+1}$; (ii) we add a new node, $x_{n+2} = -8$, and connect it with bidirectional edges to $x_0$ and $x_{n+1}$; and (iii) we set the weight of every edge to be $\frac{1}{3}$. We denote by $G$ the original graph from Sec. A.2, and by $G'$ the graph with the above changes applied. By construction, $G'$ is a circular graph with $n + 3$ nodes; hence, its diameter is $\lfloor \frac{n+3}{2} \rfloor$.

We claim that the changes made to $G'$ do not affect the behavior of nodes, i.e., in both $G$ and $G'$, in each round $t$, the same nodes move and in the same manner. Note that our construction in Sec. A.2 contains 'modules' of size 3 that are connected sequentially, together with start and end nodes, to

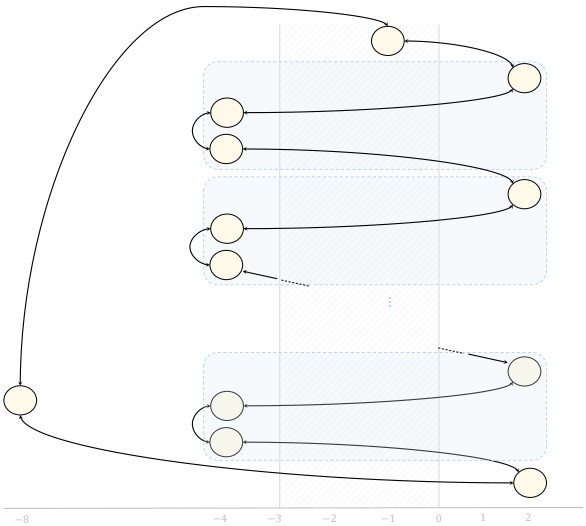

Figure 10: Circular cascading example. The diameter is $\lfloor \frac{n+3}{2} \rfloor$, but the number of movement rounds is $n$.

form a graph of size $n + 2$. To show that the node we add in this construction does not cause the final nodes to move, it suffices to consider whether $x_{n+1}$ can cross—we will show that it cannot, which implies that $x_{n+2}$ and $x_0$ will not cross either.

In the first round, we saw that in $G$, $x_1$ moves, but for all $2 \leq i \leq n$, $x_i$ does not move. Since there is no change in the features of these nodes or the edge weights, this behavior also occurs in $G'$. Note $x_0$ does not move at the first round:

$$\widetilde{w}_{0,0}(x_0^{(0)} + 3) + \widetilde{w}_{1,0}x_1^{(0)} + \widetilde{w}_{n+2,0}x_{n+2}^{(0)} = \frac{1}{3}(-1+3) + \frac{1}{3}2 + \frac{1}{3}(-8) = -\frac{4}{3} < 0$$

Next, we show that $x_{n+2}$ does not move in the first round either.

$$\widetilde{w}_{0,n+2}x_0^{(0)} + \widetilde{w}_{n+2,n+2}(x_{n+2}^{(0)} + 3) + \widetilde{w}_{n+1,n+2}x_{n+1}^{(0)} \leq \frac{1}{3}(-1) + \frac{1}{3}(-8+3) + \frac{1}{3}2 = -\frac{4}{3} < 0$$

Next, we show that $x_{n+1}$ does not move:

$$\widetilde{w}_{n+2,n+1}x_{n+2}^{(0)} + \widetilde{w}_{n+1,n+1}(x_{n+1}^{(0)} + 3) + \widetilde{w}_{n,n+1}x_n^{(0)} \leq \frac{1}{3}(-8) + \frac{1}{3}(2+3) + \frac{1}{3}(-4) = -\frac{7}{3} < 0$$

In the second round, a node can potentially move only if one of its neighbours moved in the first round. As in $G$, $x_2$ moves in $G'$ as well. We now show that $x_0$ does not move:

$$\widetilde{w}_{0,0}(x_0^{(1)} + 3) + \widetilde{w}_{1,0}x_1^{(1)} + \widetilde{w}_{n+2,0}x_{n+2}^{(1)} = \frac{1}{3}(-1+3) + \frac{1}{3}5 + \frac{1}{3}(-8) = -\frac{1}{3} < 0$$

For each round $t$, let $3 \leq t \leq n$ correspond to the node $x_t$ which moves in $G'$ as it moves in $G$. We now show that in round $n + 1$ there is no movement. In round $n$, $x_n$ is the only one that moves. Thus, the only nodes that might move are its neighbors, $x_{n-1}$ and $x_{n+1}$. Since $x_{n-1}$ has already moved at round $n - 1$, it will not move again (Prop. 1). This leaves $x_{n+1}$, which we now show does not move either:

$$\widetilde{w}_{n+2,n+1}x_{n+2}^{(n)} + \widetilde{w}_{n+1,n+1}(x_{n+1}^{(n)} + 3) + \widetilde{w}_{n,n+1}x_n^{(n)} \leq \frac{1}{3}(-8) + \frac{1}{3}(2+3) + \frac{1}{3}(-1) = -\frac{4}{3} < 0$$

This concludes that while the diameter of the graph is $\lfloor \frac{n+3}{2} \rfloor$, the number of rounds in $n$. □

