# OpenReview forum: "Strategic Classification with Graph Neural Networks"
_ICLR.cc/2023/Conference — ICLR 2023 poster_

### Official Review · Reviewer_uqWQ · 2022-10-22

**Confidence:** 3
**Correctness:** 2
**Technical Novelty And Significance:** 2
**Empirical Novelty And Significance:** 2
**Recommendation:** 3

**Clarity, Quality, Novelty And Reproducibility:**

Overall the paper is well written and easy to follow. It studied a new problem, however, the approach is only marginally novel.

**Strength And Weaknesses:**

Strength
1. The authors study an interesting question to generalize the i.i.d assumption in strategic classification to graph neural networks.

Weakness
1. Several assumptions in the problem definition are too simplistic for practical usage. On the model side: (1) it assumes nodes can directly change its embedding instead of the features which is mostly not the case. (2) The assumption of linearity and only considering direct neighbors severely limits the expressibility of GNN. On the strategy side, though, assuming access to h is already a strong assumption. The access to features of all neighbors nodes is way too strong to be practical especially in multi-step updates. Also, the assumption that all nodes have the same incentive to move towards the same class is also limiting.
2. The analysis framework of multi-step is not appropriate for the problem. It is much more natural to consider game-theoretical concepts like equilibrium. Several results from the analysis are straightforward. It would be very interesting to see results like convergence results to equilibrium etc. Also, the assumption to set gain as 2 if flipped is also arbitrary.
3. The empirical evaluation uses different assumptions compared to previous model and analysis. Section 2 allows nodes to change features arbitrarily as long as it is smaller than the gain. However, Section 5.2 assumes a max distance which induces quite different behavior.


**Summary Of The Paper:**

The authors consider the problem of strategic classification on graphs under a simplified linear graph model. The authors consider an iterative strategy for nodes changing features. An approximation learning algorithm is proposed to learn a robust classifier. The authors experiment on both synthetic 1d case and three real-world graph dataset.

**Summary Of The Review:**

Though the paper studies an interesting problem, several assumptions are too simplistic and not practical.

---

> ### Author Response · Authors · 2022-11-19
> **Response to uqWQ (part 1/2)**
>
> Thank you for your comments. To the best of our understanding, your concerns can be classified into:
> 1. **Misunderstandings:** e.g., that we assume users update embeddings and not features (we do assume they update features); that we do not consider equilibrium (our proposed dynamics do converge; this is proved in Prop. 1); that gains are arbitrary (they are fully general); or that our model of user responses is inconsistent throughout the paper (it is consistent).
> 2. **Concerns regarding assumptions:** our stated goal was to take the minimal first step from strategic classification towards graph-based learning; most of the assumptions you flag are standard in strategic classification, and are sensible for our initial efforts (see below).
> 3. **Alternative suggestions:** namely to consider “equilibria”; assuming that by this you mean *simultaneously play* rather than our choice of sequential play (which, as noted, does in fact converge to equilibrium), we argue that this alternative requires assumptions that are **much** stronger than ours.
>
> Please find below our detailed response to each of your assertions. Overall, we feel that perhaps your perspective on our work differs from ours (this we believe is expressed in your concerns regarding “practicality”). We wish to propose an alternative perspective, which is that our work is part of an ongoing effort to *make* strategic classification more practical. Strategic classification is a young literature, and carries potential for significant social impact - hence, such efforts must be approached gradually and mindfully. Our contribution lies in relaxing the key assumption of independent responses by considering interdependent classifiers, i.e., that operate on graphs; properly evaluating this step necessitates keeping as many of the other assumptions in place - which explains and justifies our modeling choices throughout.
> Our hopes are that given this new perspective, together with our detailed response, you would be willing to reconsider your evaluation.
>
> 1. **“It assumes nodes can directly change its embedding instead of the features which is mostly not the case”**
> On the contrary - we explicitly state that nodes can only change their features (Sec. 2, see paragraph “Strategic inputs”). Node embeddings are entirely susceptible to changes in the features of neighboring nodes - this is fundamental to our formulation.
>
> 2. **“The assumption of linearity and only considering direct neighbors severely limits the expressibility of GNN”**
> We respectfully disagree. Regarding linearity, from the perspective of GNNs, as we write, there is evidence suggesting that linear GNNs are sufficiently expressive to achieve performance that is on par with more complex architectures on several key tasks (see Wu et al. (2019)). As our results demonstrate, for our purposes - linear GNNs are sufficiently expressive: they achieve improved accuracy (by using the graph) in the conventional non-strategic setting; are susceptible to strategic manipulation when learned naively in  strategic setting; and allow to recover the gap (to some extent) using our proposed learning framework.
> From the perspective of strategic classification, as we write, virtually all works in this field consider linear models. One reason is that this ensures that the response function $\Delta$ is tractable; more complex classifiers can break tractability, and necessitate making "behavioral" assumptions, which have been largely unexplored, and which we wish to avoid.
> As for considering only direct neighbors: please note that we also experiment on larger neighborhoods of up to distance $K=3$ (see Appendix D.2), which for real datasets cover a considerable proportion of the graph (i.e., `small world’ effects).
> More generally, we note that our goal in this work was to take the first, minimal necessary step from standard strategic classification and towards graph-based learning. Remaining within the regime of linear classifiers achieves this, and allows us to make connections and draw conclusions that relate to current literature. Given that our results demonstrate that linear GNNs still produce complex phenomena when coupled with strategic behavior, and in ways that lead to challenging learning problems - we do not see a clear argument for why our modeling choice is insufficiently expressive.
>
> 3. **“Assuming access to h is already a strong assumption”**
> Yes, but it is also fundamental to strategic classification in general. Some current works in this space aim to relax this assumption (see footnote 6). However, one conclusion that clearly arises is that doing so is non-trivial. For example, Ghalme et al. (2021) argue that if the system does not reveal $h$, it also loses its ability to anticipate user responses; in this case, it is unclear how user responses should be modeled, nor is it clear how the system should utilize its control over $h$.

---

> > ### Author Response · Authors · 2022-11-19
> > **Response to uqWQ (part 2/2)**
> >
> > 4. **“The access to features of all neighbors nodes is way too strong to be practical especially in multi-step updates”**
> > Here as well we respectfully disagree. There are many social network applications in which user attributes are either publicly visible; in others, attributes are revealed to network friends. In our formulation, viewing access to neighboring features is sufficient for performing strategic updates.
> >
> > 5. **“Also, the assumption that all nodes have the same incentive to move towards the same class is also limiting.”**
> > This is an interesting point. First, we argue that in some applications it is quite reasonable to assume that all users gain from one of two possible decision outcomes; this holds for all canonical applications of strategic classification, such as loans, admissions, and hiring. However, we also agree that this does not cover all applications. In some settings, different users may want different outcomes; the question is then - what should be the target of prediction (i.e., $y$)? A recent paper by Levanon & Rosenfeld (2021) studies an incentive-aligned setting where users also want accurate predictions; this is an interesting avenue to approach, but requires modeling users as having access to additional information, and is therefore beyond our scope. Beyond this, we are not familiar with any other feasible alternatives that have been proposed or studied.
> >
> > 6. **“The analysis framework of multi-step is not appropriate for the problem. It is much more natural to consider game-theoretical concepts like equilibrium. It would be very interesting to see results like convergence results to equilibrium etc.”**
> > This comment puzzles us: quite explicitly, our work **does** consider equilibrium - this is exactly the statement we make in Corr. 1 (based on Prop. 1), which shows that our proposed response model generates dynamics that are guaranteed to converge to equilibrium (i.e., no user has incentive to unilaterally move). We rely on this property in our learning approach in Sec. 4, and study it empirically in Sec. 5.
> > This leaves us to ask if by “equilibrium” the reviewer means that it would have been preferable to model users as acting concurrently, i.e., modeling strategic behavior as a simultaneous game rather than a dynamic game (as we do). If this is indeed the case - then we fail to see the logic behind this reasoning. Simultaneous games with pure strategies do not necessarily have equilibrium; and if they do, more often than not, there is no unique equilibrium (please see our concrete response to Rev. rHAu). How would you then approach learning? And even if there is a unique equilibrium - why would it be computable? (And nevertheless differentiable, if we were to use gradient methods for optimization).
> > Conversely, the dynamics we propose - which we believe are a natural extension of the conventional response model in vanilla strategic classification - converge to a unique equilibrium that is computable. Dynamics  - and in particular best-response dynamics - are a standard construct in economic modeling. Converging dynamics often serve to justify discussion of equilibria, and equilibria that can be shown to be the product of converging dynamics are often viewed favorably (e.g., the *tatonnement* principle in market theory). Finally, we note that simultaneous plays necessitates assuming that users act by reasoning about the potential behavior of all other players in the game. We feel this assumption is considerably stronger than those we make (e.g., vs. assuming users have access to neighboring features). This strikes us as somewhat contradictory to some of the earlier comments regarding our assumptions.
> >
> > 7. **“The assumption to set gain as 2 if flipped is also arbitrary.”**
> > Yes, but it nonetheless permits full generality. Assuming user utility derives from $h(x)$ (and is therefore either -1 or +1) is conventional. The choice of $\pm 1$ as values that utility takes therefore merely reflects a choice of scale. Since costs can also be scaled, and since the argmax is invariant to an additive constant, this choice does not restrict the model in any sense (e.g., we could have $y \in \{0,1\}$, or any other values).
> >
> > 8. **“The empirical evaluation uses different assumptions compared to previous model and analysis. Section 2 allows nodes to change features arbitrarily as long as it is smaller than the gain. However, Section 5.2 assumes a max distance which induces quite different behavior.”**
> > We assure the reviewer that Sec. 2 and Sec. 5 are entirely consistent, and that throughout our paper, users are modeled as responding via Eq. (4). In Sec. 5.2, $d_\beta = 2 / \beta$ reflects the maximal possible movement distance - it does not imply that updates necessarily move this distance.

---

> ### Author Response · Authors · 2022-12-11
> **Response to uqWQ**
>
> Dear reviewer  uqWQ,
>
> We hope our responses were helpful in adequately addressing your earlier concerns.
> In case you have any further questions or comments, please let us know, and we will gladly respond.

---

### Official Review · Reviewer_KAMH · 2022-10-23

**Confidence:** 2
**Correctness:** 4
**Technical Novelty And Significance:** 2
**Empirical Novelty And Significance:** 2
**Recommendation:** 6

**Clarity, Quality, Novelty And Reproducibility:**

There is a github repo associated with the submission; so it is believable that the reported results can be reproduced by following the documentation there.

Strategic classification is related to the study of game theory; in particular the proposed formulation of this work is very much inspired by an ITCS paper by Hardt et al. (2016). This paper follows the formulation of this paper and applies concepts such as myopic response in graph neural networks. From this perspective, this paper seems like a natural extension of Hard et al. (2016) to the context of GNNs.

**Strength And Weaknesses:**

This paper is generally well-written; the methods and the experiments are both clearly described. The experiments also provide clear evidence about the proposed method in the presence of strategic behaviors of users.

One question with the reported results (e.g., in table 1) is to what extent is the gap between robust and non-strategic performance necessary. Whereas for CiteSeer, there is only a 2% gap between the two settings; for Cora and PubMed, there are about 10% gap between the two settings. To what extent can we close the performance gap between the robust performance and the non-strategic performance?

Another question is to what extend do these results hold under architectural modifications—e.g., suppose one changes the diffusion matrix in the GNN, would the same result still apply?

**Summary Of The Paper:**

This paper studies strategic classification under graph neural networks. In the model of strategic user behavior, it is assumed that a user plays myopic best-response over a sequence of multiple update rounds. The myopic response mapping means that a user may move their reported features in the direction that best improves their classification outcome from the graph neural network model. Several results are then stated to demonstrate the basic properties of the myopic response model and the consequent dynamics.

Then, the learning objective based on the myopic best response is optimized over multiple rounds. This optimization computes the approximately modified features after T rounds.

Empirical evaluation is provided to demonstrate different aspects of learning in the above setting. Several experiments are conducted on popular settings in the GNN literature, this includes the Cora, CiteSeer, and PubMed datasets, trained with a simplified GCN architecture. The results show that by anticipating the strategic behavior of users in the GNN, the optimized graph neural network performs robustly against the naive baseline, which trains a GNN without accounting for any strategic behaviors.

**Summary Of The Review:**

Overall this paper studies strategic classification in graph neural networks; it adopts a myopic best response strategy for modeling user’s strategic behavior. Then it incorporates such myopic behavior into an optimization procedure. Experiments verify the robustness of this optimization method. On the other hand, it remains unclear to what extend could one close the gap between robust and non-robust performance, and how general the reported result is (in the broader context of GNN literature).

---

> ### Author Response · Authors · 2022-11-19
> **Response to KAMH**
>
> Thank you for your response and insightful questions, which we answer below.
>
> 1. **“To what extent is the gap between robust and non-strategic performance necessary? To what extent can we close it?”**
> Thank you for this question. Broadly, whether a gap exists - as well as to what degree it is reducible - greatly depends on the topology of the graph and the underlying joint distribution of features and labels. This has also been observed in adversarial learning on graphs when employing defensive strategies (e.g., [1,2], which include the datasets we use). Gaps can also result from the use of proxy objectives, but in a way that also depends on the graph, e.g., due to over-smoothing manifesting differently on different graph topologies [3].
>
> Note that sometimes subtle changes can have dramatic impact: we have now added to Appendix E.1 a simple example in which the baseline achieves 100% accuracy, and it is possible to obtain a gap of 0% (i.e., recover perfect accuracy); however, a slight modification to the features of one node causes the best obtainable gap to jump to a considerable 33% (i.e., after this change, no classifier can achieve more than 67% accuracy). This suggests that in practice it can be quite hard to anticipate in advance what gaps to expect.
>
>
> 2. **“To what extent do these results hold under architectural modifications?”**
> Please see Appendix D.2, which includes an experiment that examines outcomes under different architectural choices (while remaining within the bounds of our general modeling choices). In particular, we vary the number of propagation layers K; results show that results are fairly robust to this change, and that there is no clear trend across datasets as K increases.
> We note that we've also experimented with different normalizations of the graph Laplacian (another parameter of the neural architecture), but these seemed to make very little difference in terms of experimental outcomes, and so were not included.
>
>
> [1] Robustness of Graph Neural Networks at Scale (Neurips 2021)
>
> [2] Not All Low-Pass Filters are Robust in Graph Convolutional Networks (Neurips 2021)
>
> [3] On the Bottleneck of Graph Neural Networks and its Practical Implications (ICLR 2021)

---

> ### Author Response · Authors · 2022-12-12
> **Response to KAMH**
>
> Dear reviewer KAMH,
>
> We hope our responses were helpful in adequately addressing your earlier concerns. In case you have any further questions or comments, please let us know, and we will gladly respond.

---

### Official Review · Reviewer_aLDo · 2022-10-24

**Confidence:** 2
**Correctness:** 4
**Technical Novelty And Significance:** 3
**Empirical Novelty And Significance:** 3
**Recommendation:** 8

**Clarity, Quality, Novelty And Reproducibility:**

This paper is well-written and easy to follow. This paper proposes a novel strategic classification on graph setting and a new algorithm to learn a robust classifier. I think their results are original and reproducible.

**Strength And Weaknesses:**

Strengths:
1. This paper proposes a novel strategic classification setting on the graph.
2. This paper analyzes the strategic behavior of users in interaction with others.
3. They propose a differentiable framework to learn a robust classifier in the strategic classification on graph setting.

Weaknesses:
1. It seems properties in section 3.2 and the learning algorithm highly depend on that the classifier is linear. I understand that linear classifier is widely used in practice and more accessible as the first step toward this problem.
2. At the beginning of section 3 and experiments, you normalize the weights such that the node embedding depends on one parameter $\alpha$. It seems this simplifies the effects of the graph structure and makes the problem easier.

**Summary Of The Paper:**

This paper considers the strategic classification problem on graphs. Specifically, they consider a linear graph-based classifier, which classifies each node in the graph based on the node embedding with a linear function. The node embedding of each node depends on the features of this node and also the features of its neighborhood. In the strategic classification setting, each user (node) can modify the features of this node to change the classification result. The strategic classification problem is to learn a linear graph-based classifier to minimize the misclassification loss under strategic behavior.

This paper shows that the inter-user dependency on the graph might be exploited by strategic users and cause a higher loss. They also propose a differential framework for strategically-robust learning of graph-based classifiers.  Their theoretical results and method are supported by experimental results.

**Summary Of The Review:**

I think this paper is interesting and provides valuable insights into the strategic classification with interaction, which might be useful for further investigation.

---

> ### Author Response · Authors · 2022-11-19
> **Response to aLDo**
>
> Thank you for your positive review!
>
> 1. **“It seems properties in section 3.2 and the learning algorithm highly depend on that the classifier is linear. I understand that linear classifiers are widely used in practice and more accessible as the first step toward this problem.”**
> Indeed - linear classifiers are sufficiently expressive for our purposes, permit tractable user responses, and allow us to make connections to both the strategic classification and GNN literatures.
>
> 2. **“At the beginning of section 3 and experiments, you normalize the weights such that the node embedding depends on one parameter $\alpha$. It seems this simplifies the effects of the graph structure and makes the problem easier.”**
> Please note that Eq. (3) in Sec. 3 defines *individualized* $\alpha_i$. For normalized weights, which are common in GNNs, these permit full generality, and are used simply as convenient notation. To see why consider that for any normalized weights $w_\{ii\}$ and {$w_\{ij\}$}$_j$, we can set $\alpha_i=1-w_\{ii\}$ and define $w’_\{ij\}=w_\{ij\} / \alpha_i$, so that $\bar\{x\}_i = \sum_\{j \neq i\} w’_\{ji\} x_j$, which gives Eq. (3).  For unnormalized weights, $\alpha_i$ can be scaled according to the sum of weights, $\sum_j w_\{ji\}$. Thus, **all of our results hold for general weights**. The only exception is Sec. 5.1, in which we intentionally and statedly use a global $\alpha$ (but nonetheless still use individualized weights) for experimental purposes.

---

### Official Review · Reviewer_rHAu · 2022-10-26

**Confidence:** 3
**Correctness:** 4
**Technical Novelty And Significance:** 3
**Empirical Novelty And Significance:** 3
**Recommendation:** 8

**Clarity, Quality, Novelty And Reproducibility:**

Main ideas of the paper is easy to follow without much background on strategic classification. Authors do a great job providing high-level insights on both mathematical results and empirical results. The analysis seems relatively simple because users are assumed to make myopic decisions, but still the convergence result provides some insight. As mentioned above, connections to graph properties could've made these analysis much more interesting.

The formulation seems to be novel, significantly departing from regular assumptions in the topic of strategic classification. The formulation seems to be applicable to many extensions to practical problems.

Experiments of this paper seems to be reproducible, as authors share code, and experiments use public data only.

**Details Of Ethics Concerns:**

Authors do a good job describing ethics and social implications. The use of GNNs and strategic behavior of users may have societal implications, but at this point the model is a bit simplistic for results of this paper to actually impact users' behavior on real-world systems.

**Strength And Weaknesses:**

The problem formulation is elegant. It is different enough from existing strategic classification formulations to introduce interesting dynamics across users. It is realistic enough to motivate real-world scenarios like credit scoring for microlending. It is simple enough to allow interesting mathematical analysis and possible to be adapted for more complex problems. Authors made a good judgment balancing these considerations.

Both mathematical analysis and empirical analysis provide good insights into how GNN introduces dynamics to the collective behavior of users. On the other hand, neither mathematical analysis nor empirical analysis actually take the property of the graph into consideration. This is unfortunate, because for sure the characteristics of graphs would play a crucial role determining the dynamics. For example, if the graph is divided into components, no information shall propagate across components. Short-radius graphs would propagate changes faster than long-radius graphs. A central node (say, in terms of PageRank or other centrality metrics) would play a bigger role impacting the dynamics than peripheral nodes would.

The fact that users only make myopic decisions also make the problem less interesting. Users would not really take advantage of opportunities to "collude" with their neighbors. Given it is reasonable to assume users shall communicate their plans with their social network, the fact that this works stops at myopic model is disappointing.

**Summary Of The Paper:**

Authors formulate the problem of strategic classification on graphs, which users shall modify their features with some cost (strategic classification) and the predictor on a user relies on the smoothed embedding of neighbors of a user. Users are assumed to make myopic decisions based on other users' previous behavior. The fact that one user's actions can impact predictions on other users introduces interesting dynamics on the collective behavior of users. Authors identify interesting properties of the dynamics with both mathematical analysis and numerical experiments.

**Summary Of The Review:**

Both practically & analytically interesting problem formulation with good insights from both analytical & numerical experiments. However the analysis could've been strengthened by taking graph properties into account, or considering non-myopic behavior of users.

---

> ### Author Response · Authors · 2022-11-19
> **Response to rHAu part (1/2)**
>
> Thank you for your comments and suggestions; we appreciate your reviewing efforts. Please see our answers to your inquiries below.
>
> 1. **“Connections to graph properties could've made these analyses much more interesting”**
> Thank you for your suggestion - in response, we’ve added to the paper several new results which shed some light on how graph structure and node properties can affect outcomes.
> These include:
> * Analytic #1: A negative result on the spread of influence and graph diameter.
> * Analytic #2: For a given clique, either all nodes move, or none do.
> * Experimental: We empirically examine the influence of nodes with high degree centrality
>
> These new results now appear in the revised Appendix D.3 (experimental) and appendix E (analytic).
>
> We would, however, like to make a general comment to align expectations. As we’ve shown in other aspects of our proposed formulation, strategic behavior in graphs can result in complex, often unintuitive outcomes. This holds also in regards to graph structure and node properties - which makes it quite challenging to produce general results regarding the role of graph structure or node properties.
> For example, consider a central node $i$ of high out-degree (and for simplicity assume uniform edge weights). As you note, intuitively we would expect such a node to be influential. In practice, however, this non-trivially depends on other aspects of the setup:
> * If $i$ is too far from the decision boundary, it will have no positive influence
> * If $i$ is close enough to move, and moves exactly on the decision boundary - it will still have no positive influence
> * Conversely, a peripheral node $j$ can influence *all* nodes (see Appendix A.1).
> * To have some positive influence, $i$ *must* have in-neighbors that are classified as negative, and are sufficiently far from the decision boundary (similarly to case (B) in Fig. 1)
> * There are cases in which the distinction between (i) $i$ preventing all of his out-neighbors from crossing, and (ii) $i$ pulling all of his neighbors over the boundary -  is an $\epsilon$-difference in the position of $x_i$
> * Holding all else fixed, another low-degree node $j$ can be much more influential than the high-degree $i$ if $j$’s out-neighbors have lower in-degrees than $i$’s out-neighbors (but not necessarily)
> * Holding all else fixed, another high-degree node $j$ can be more influential than $i$ if its in-degree is higher than $i$’s, i.e., if $j$ is in itself more susceptible to external influence (but not necessarily)
> These exemplify why making general claims about the relation between graph properties and outcomes (here, degree centrality and influence) is perhaps appealing, but highly non-trivial. Our hope is that our new results, combined with our original efforts, will be considered under this nuanced perspective.
>
> 2. **“The analysis could've been strengthened by … considering non-myopic behavior of users”**
> Thank you for this suggestion as well. While we agree that modeling user behavior beyond myopic is interesting, doing so is far from straightforward. Consider the following simple example: Let $i,j$ be two nodes connected by a bi-directional edge, with one-dimensional features $x_i=x_j=-1$. Let $h$ be a classifier that thresholds at 0, and for simplicity, assume $h(0)=-1$.
> Now consider incentives and movement: Independently, each of the two nodes can cross; for example, if $j$ is kept fixed, then $i$ can cross by moving to $x’_i=1$ (note it must move beyond the decision boundary since it is  "held back" by the edge $(j,i)$) - this gives $\phi_i=0$ and so $\hat{y}_i=1$. Note this means that also $\hat{y}_j=1$, i..e, $j$ crosses for free (whereas $i$ invested $c(x_i,x’_i)=2$). In other words, $j$ is *free riding* on $i$’s efforts. From $i$’s perspective, this is suboptimal (and unfair): due to symmetry, in the same manner, $j$ could have been the one to invest the necessary effort to permit them both to cross. If $i$ and $j$ are allowed to coordinate, they might agree to jointly move exactly on the decision boundary, $x’_i=x’_j=0$, so that $\phi_i=\phi_j=0$ and hence $\hat{y}_i=\hat{y}_j=1$. This, however, is not optimal for either $i$ nor $j$ - both would prefer to free-ride, and would be tempted to revert back to $x=-1$ to `force’ the other into moving unilaterally. If both nodes are modeled as moving simultaneously, then this results in a continuous variation on the game of chicken, which is known to have three distinct equilibria; from a learning perspective, it is unclear how to determine a-priori which of these equilibria learning should target (in particular given that different instantiations induce different incentive structures).

---

> > ### Author Response · Authors · 2022-11-19
> > **Response to rHAu part (2/2)**
> >
> > These considerations pertain to a graph having two nodes and one edge; reasoning about the strategic outcomes under general graphs is considerably more challenging (see, e.g., [1]). And while there exists a body of work on games played over graphs or networks [2], to the best of our knowledge these do not relate to learning scenarios, nor do they include a learning entity as one of the agents in the game.
> >
> > Overall, our aim in this paper was to take the first and minimally-necessary step beyond strategic classification and towards graph-based learning. Our modeling choices -  which we have thought long and hard about - were made accordingly, and with the intent of achieving a balanced problem setup (which we were happy to learn that you viewed favorably). Our work set out to relax the assumption of independent user responses; our results show that this can be achieved **even when responses remain myopic**, and in a way which results in complex (and even surprising) behavior. As our example above illustrates, considering more elaborate forms of user behavior is an intriguing challenge - but one we feel is deserving of independent future efforts.
> >
> > [1] Anti-coordination and social interactions (2006)
> > [2] Games on Networks (2014)

---

> > > ### Comment · Reviewer_rHAu · 2022-12-05
> > > **thanks for updates**
> > >
> > > Thanks for updates. Most of my concerns are addressed in these follow-up analysis, so I increased my score accordingly.

---

> ### Author Response · Authors · 2022-12-11
> **Thank you for increasing your score**
>
> Dear reviewer  rHAu,
>
> Thank you for increasing your score! We were happy to learn that you found our response to be helpful.

---

### Author Response · Authors · 2022-11-19
**Response to all reviewers**

We would like to thank all reviewers for their time and effort, as well as for raising important issues and questions regarding our work. As per your requests, we have added to the revised manuscripts additional results, both analytic and experimental. These include:

* Analytic #1: A negative result on the spread of influence and graph diameter.
* Analytic #2: For a given clique, we prove that either all nodes move, or none do.
* Analytic #3: An example showing that a large gap can exist between strategic and non-strategic performance—but which closes completely when a mild variation is introduced.
* Experimental: An empirical analysis of the influence of high-degree nodes.

Please find below our detailed answers to each individual reviewer’s questions and comments.

---

### Decision · Program_Chairs · 2023-01-20

**Decision:**

Accept: poster

**Justification For Why Not Higher Score:**

The results could be strengthened by taking graph properties into account and/or considering non-myopic behavior of users.

**Justification For Why Not Lower Score:**

This is one of the first papers addressing strategic classification in graphs. Interesting analysis/methodology and strong experiments.

**Metareview: Summary, Strengths And Weaknesses:**

This paper introduces the problem of strategic classification on graphs. More specifically, they consider a linear graph-based classifier, which classifies each node in the graph based on the node embedding with a linear function. Each user (node) can modify their features of to change the classification result. The strategic classification problem is to learn a linear graph-based classifier to minimize the misclassification loss under strategic behavior.

A majority of the reviewers appreciated the contribution and were in support of accepting the paper. However, they did bring up a number of points for improvement, which the authors should address in the final version of their paper.

**Note From Pc:**

if the above contains the word "oral" or "spotlight" please see: "oral" presentation means -> notable-top-5% and "spotlight" means -> notable-top-25%. As stated in our emails, we are disassociating presentation type from AC recommendations